# The switch-like expression of heme-regulated kinase 1 mediates neuronal proteostasis following proteasome inhibition

**Beatriz Alvarez-Castelao[†], Susanne tom Dieck, Claudia M Fusco, Paul Donlin-Asp, Julio D Perez, Erin M Schuman***

Max Planck Institute for Brain Research, Frankfurt am Main, Germany

**Abstract** We examined the feedback between the major protein degradation pathway, the ubiquitin-proteasome system (UPS), and protein synthesis in rat and mouse neurons. When protein degradation was inhibited, we observed a coordinate dramatic reduction in nascent protein synthesis in neuronal cell bodies and dendrites. The mechanism for translation inhibition involved the phosphorylation of eIF2$\alpha$, surprisingly mediated by eIF2$\alpha$ kinase 1, or heme-regulated kinase inhibitor (HRI). Under basal conditions, neuronal expression of HRI is barely detectable. Following proteasome inhibition, HRI protein levels increase owing to stabilization of HRI and enhanced translation, likely via the increased availability of tRNAs for its rare codons. Once expressed, HRI is constitutively active in neurons because endogenous heme levels are so low; HRI activity results in eIF2$\alpha$ phosphorylation and the resulting inhibition of translation. These data demonstrate a novel role for neuronal HRI that senses and responds to compromised function of the proteasome to restore proteostasis.

**\*For correspondence:**
erin.schuman@brain.mpg.de

**Present address:** [†] Department of Biochemistry and Molecular Biology, Veterinary School, Complutense University of Madrid, Madrid, Spain

**Competing interests:** The authors declare that no competing interests exist.

## Introduction

The concept of an optimal protein concentration and its associated regulatory mechanisms is known as 'proteostasis'. While many studies have focused on proteostatic regulators that are associated with the proper synthesis, folding and elimination of protein aggregation (*Klaips et al., 2018*), less attention has been paid to the coordination of protein synthesis and degradation within cells. The study of proteostasis is particularly important in neurons because alterations in protein synthesis and degradation, particularly via the ubiquitin proteasome system (UPS), have emerged as network hubs in many disease states (*Ciechanover and Brundin, 2003*; *Labbadia and Morimoto, 2015*; *Tai and Schuman, 2008*).

As an additional challenge, neurons must dynamically regulate the synaptic proteome in response to plasticity elicited by neural activity. Many studies have demonstrated that de novo protein synthesis plays an important role in synaptic transmission and plasticity, (*Holt et al., 2019*; *Sutton and Schuman, 2006*). In addition, protein degradation, mediated by the ubiquitin proteasome system (UPS), also plays an important role in regulating synaptic function (*Bingol and Schuman, 2006*; *Tai and Schuman, 2008*; *Ramachandran and Margolis, 2017*). In neurons, proteome regulation occurs in the cell body as well as the dendrites and axons allowing the initiation of proteome remodeling at synapses, far from the cell body (*Holt et al., 2019*). While it is clear that changes in synaptic transmission involve extensive regulation of the synaptic proteome via the regulated synthesis and degradation of proteins, it is not well understood how these two processes are coordinately regulated to achieve the desired level of individual proteins at synapses.

To address this question, we studied the impact of proteasome inhibition on protein synthesis in mature neurons. We found that blocking proteasome function leads to a coordinate reduction in protein synthesis, indicating the existence of a global proteostatic pathway in neurons. The entry point for down-regulation of protein synthesis is the much-studied protein synthesis initiation factor, eIF2α (*Donnelly et al., 2013*). The phosphorylation of eIF2α that leads to reduced neuronal translation is accomplished by one of the eIF2α kinases, heme-regulated inhibitory kinase (HRI), best known for its translational control in erythrocyte precursors. The activity and expression of neuronal HRI is regulated in a biologically clever manner, expression is kept low by a short- half-life and a translational control mechanism that is relieved by proteasome inhibition- augmenting HRI expression and leading to the coordinate regulation of protein synthesis.

## Results

### Neuronal proteasome inhibition leads to a coordinate reduction in neuronal protein synthesis, globally and locally in dendrites

To determine whether the processes of neuronal protein synthesis and degradation are coupled, we examined the effect of blocking proteasomal degradation on nascent protein synthesis in cultured hippocampal neurons (*Figure 1A*). Blocking proteasome activity with either of two chemically distinct inhibitors (MG-132, 10 µM; Lactacystin, 10 µM; *Figure 1A–E* and *Figure 1—figure supplement 1A–F*) for 2 hr led to a dramatic decrease in protein synthesis, measured with two different labeling methods: the non-canonical amino acid azidohomoalanine (AHA) and BONCAT (biorthogonal non-canonical amino acid tagging) (*Dieterich et al., 2006*; *Figure 1B–C*) or metabolic labeling with puromycin (*Figure 1D,E* and *Figure 1—figure supplement 1E*). The compensatory proteostatic decrease in protein synthesis did not require a complete block of proteasomal activity: dose-dependent reductions in proteasome activity led to coordinate reductions in protein synthesis (*Figure 1D* and *Figure 1—figure supplement 1F*). These data suggest that cross-talk can occur between the protein degradation and synthesis pathways in response to relatively small fluctuations in proteasome activity.

Because cultured brain preparations contain a mix of neuronal and glial cell types, we also visualized protein synthesis, in situ, using puromycylation to detect protein synthesis (*Schmidt et al., 2009*). In neurons, we observed that proteasome inhibition led to a decrease in protein synthesis apparent in both the cell bodies and dendrites of neurons (*Figure 1E*). The reduction in protein synthesis was not due to cell death (*Figure 1* and *Figure 1—figure supplement 2A*) or compromised cell health as protein synthesis levels were restored following an extensive (14 hr) washout of the proteasome inhibitor (*Figure 1* and *Figure 1—figure supplement 2B–D*). Furthermore, the overexpression of a mutant ubiquitin molecule, K48R+G76A (which inhibits polyubiquitination and is resistant to deubiquitinases thus inhibiting protein degradation by the proteasome [*Hodgins et al., 1992*; *Ju and Xie, 2004*; *Ju and Xie, 2006*] also decreased protein synthesis (*Figure 1F* and *Figure 1—figure supplement 2,E–F*).

To determine if the proteostatic mechanism also functions in neuronal processes, we used a special membrane which separates neuronal cell bodies from dendrites and axons ('neurites') (see Materials and methods). We added a proteasome inhibitor for 2 hr, then physically removed the cell bodies from the region above the membrane and performed a brief (5 min) metabolic labeling (using AHA and BONCAT) on the isolated dendrites and axons inhabiting the region below the membrane (*Figure 1G,H*). We found that protein synthesis in neurites was also significantly inhibited by proteasome inhibition (*Figure 1I,J*), indicating that the feedback mechanism coupling protein degradation to synthesis also functions locally in neuronal processes.

### Phosphorylation of the initiation factor eIF2α by HRI is responsible for reduced neuronal protein synthesis following proteasome inhibition

To determine the mechanisms by which protein synthesis is reduced, we first asked whether proteasome inhibition brings about a reduction in protein synthesis by altering transcription. Blocking transcription during proteasome inhibition, however, did not affect the reduction in protein synthesis we previously observed (*Figure 2—figure supplement 1A–B*). Because proteasome activity can contribute to amino acid recycling in cells, we measured the free amino acid content of the neurons, and

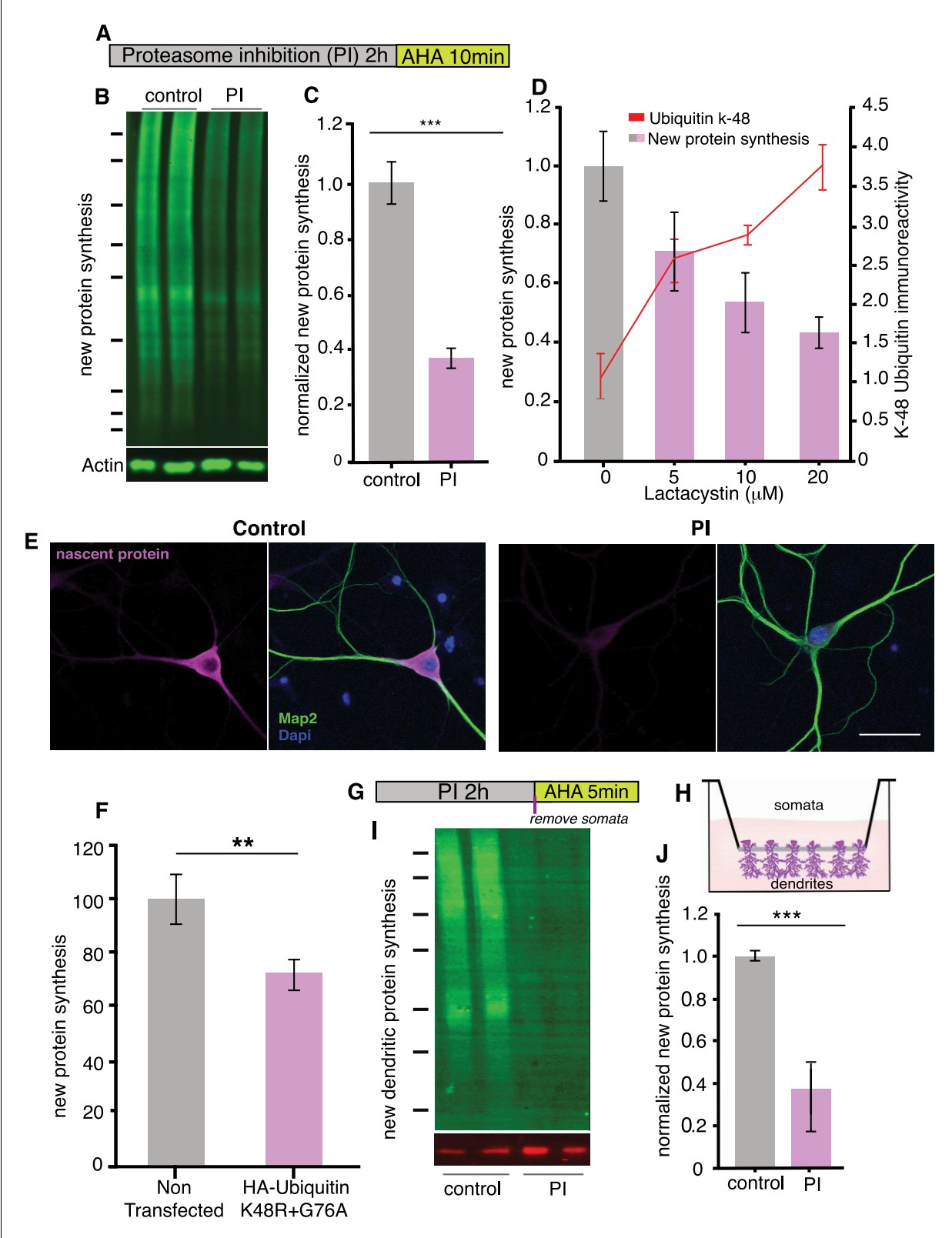

**Figure 1.** Proteasome inhibition leads to a coordinate inhibition of protein synthesis in neurons. (**A**) Scheme indicating experimental protocol: cultured hippocampal neurons were exposed to a proteasome inhibitor (PI; MG132 or lactacystin- see supplementary figs) for 2 hr and then the non-canonical amino acid AHA was added for 10 min to label newly synthesized proteins (see Materials and methods). (**B**) Representative BONCAT western blot using an anti-biotin antibody to detect newly synthesized proteins (labeled with AHA and then clicked with a biotin alkyne tag). Shown are two biological

*Figure 1 continued on next page*

*Figure 1 continued*

replicates each from a control sample (not treated with a PI) and a PI-treated sample, respectively. (C) Analysis of experiment shown in B. Protein synthesis in hippocampal neurons was significantly reduced following proteasome inhibition. p<0.001, unpaired t-test, three experiments. Error bars = SD. (D) Analysis of newly synthesized proteins (puromycylation; bar graphs) and K48 ubiquitin chains (line graph) after treatment of cultured neurons with 5, 10 or 20 μM Lactacystin. t-Test (multiple comparisons) n = 4 experiments for each concentration. For protein synthesis, 0 vs 5 μM p<0.05, 0 vs 10 μM p <0.01, 0 vs 20 μM p<0.01, For K48 ubiquitin, 0 vs 5 μM p<0.01, 0 vs 10 μM p <0.001, 0 vs 20 μM p<0.01. (E) Metabolic labeling (magenta; using puromycylation, see Materials and methods) of the global nascent protein pool in hippocampal cell bodies and dendrites following proteasome inhibition, nucleus (DAPI, blue) and dendrites (MAP2, green) can also be visualized in the images. Scale bar = 50 microns. (F) Analysis of protein synthesis (using puromycylation) of the global nascent protein pool in hippocampal neurons transfected with HA-Ubiquitin K48R+G76A –see *Figure 1—figure supplement 2*, (unpaired t-tests p<0.01, three experiments, n = 110 mock n = 105 ubi-K48R+G76A-transfected neurons). (G) Scheme of experiment to determine whether dendritic protein synthesis is also altered, neurons were treated with PI and labeled with AHA for the last 5 min. (H) Scheme of the experimental set up, neurons were cultured on a special membrane that allows dendrites and axons, but not cell bodies, to grow through pores. The proteasome was inhibited, then the cell bodies were scraped away from the top of the membrane and metabolic labeling was conducted on the dendritic fraction. (I) Inhibition of the proteasome results in a coordinate inhibition of dendritic protein synthesis. Representative BONCAT western blot showing the metabolically labeled newly synthesized dendritic protein in green. (J) Analysis of experiment shown in (I). Protein synthesis in dendrites was significantly reduced by over 60% following proteasome blockade, (p<0.001, unpaired t-test, four experiments) Error bars = SEM. Molecular weight markers in B and G from top-to-bottom are 250, 150, 100, 75, 50, 37, and 25 kD.

The online version of this article includes the following source data and figure supplement(s) for figure 1:

**Source data 1.** Source data for *Figure 1A and G*.
**Source data 2.** Source data for *Figure 1C, D, F, J*.
**Figure supplement 1.** Additional experiments examining the effects of proteasomal inhibition on neuronal protein synthesis.
**Figure supplement 1—source data 1.** Source data for *Figure 1—figure supplement 1A, B, F*.
**Figure supplement 1—source data 2.** Source data for *Figure 1—figure supplement 1C, D*.
**Figure supplement 2.** Neuronal health assessment after PI, and control experiments for ubiquitin over-expression.
**Figure supplement 2—source data 1.** Source data for *Figure 1—figure supplement 2A, D, F*.
**Figure supplement 2—source data 2.** Source data for *Figure 1—figure supplement 2C*.

we did not find any changes following proteasome inhibition (*Figure 2—figure supplement 1C*). We thus focused on protein synthesis directly and examined the phosphorylation status, following proteasome inhibition, of several canonical translation factors known to influence protein synthesis initiation and elongation rates. The translation factors eIF4B, eIF4EBP1/2, and eIF4G showed no change in phosphorylation (*Figure 2—figure supplement 1D–E*). In contrast, as has been observed in mouse embryonic fibroblasts and other cell types (*Fribley et al., 2004*; *Jiang and Wek, 2005*; *Mazroui et al., 2007*; *Nawrocki et al., 2005*; *Vabulas and Hartl, 2005*; *Yerlikaya et al., 2008*; *Zhang et al., 2010*), we observed a marked and rapid increase in the phosphorylation of eIF2α (a key regulator of translation initiation) (*Harding et al., 2000*), without a change in total eIF2α levels (*Figure 2A–C*). Consistent with a causal role of eIF2α phosphorylation in the inhibition of neuronal protein synthesis, we found that treatment with the integrated stress response inhibitor (that acts downstream of peIF2α) (ISRIB; [*Sidrauski et al., 2015*]) reversed the effects of proteasomal inhibition on protein synthesis (*Figure 2D,E*), without affecting basal levels of translation (*Figure 2—figure supplement 1F*).

There are four kinases that are known to phosphorylate eIF2α, the eIF2α kinases 1–4, commonly known as PERK (PKR-like ER kinase), PKR (protein kinase double-stranded RNA-dependent), GCN2 (general control non-derepressible-2), and HRI (heme-regulated inhibitor) (*Donnelly et al., 2013*; *Taniuchi et al., 2016*). In order to evaluate which kinase (or kinases) phosphorylates eIF2α in response to proteasome inhibition, we first focused on GCN2 since it is well-established as a regulator of translation in neurons during synaptic plasticity (*Costa-Mattioli et al., 2005*; *Trinh and Klann, 2013*), and was previously described to be activated in response to proteasome inhibitors in *D. melanogaster* (*Suraweera et al., 2012*). Using cultured neurons from GCN2 knock-out mice we examined the sensitivity of protein synthesis to proteasomal inhibition. Surprisingly, in the absence of GCN2 protein synthesis was still inhibited by proteasome blockade (*Figure 3A*). We conducted the same experiments in cultured neurons obtained from PERK knock-out mice or in PKR-inhibited neurons and again observed no effect on the proteasome-dependent inhibition of protein synthesis (*Figure 3A*). We thus turned our attention to the least likely candidate, HRI, a kinase that is primarily activated by reduced cellular heme levels and is known to play an important role in regulating globin translation in erythrocytes (*Han et al., 2001*). Using neurons from an HRI knock-out mouse

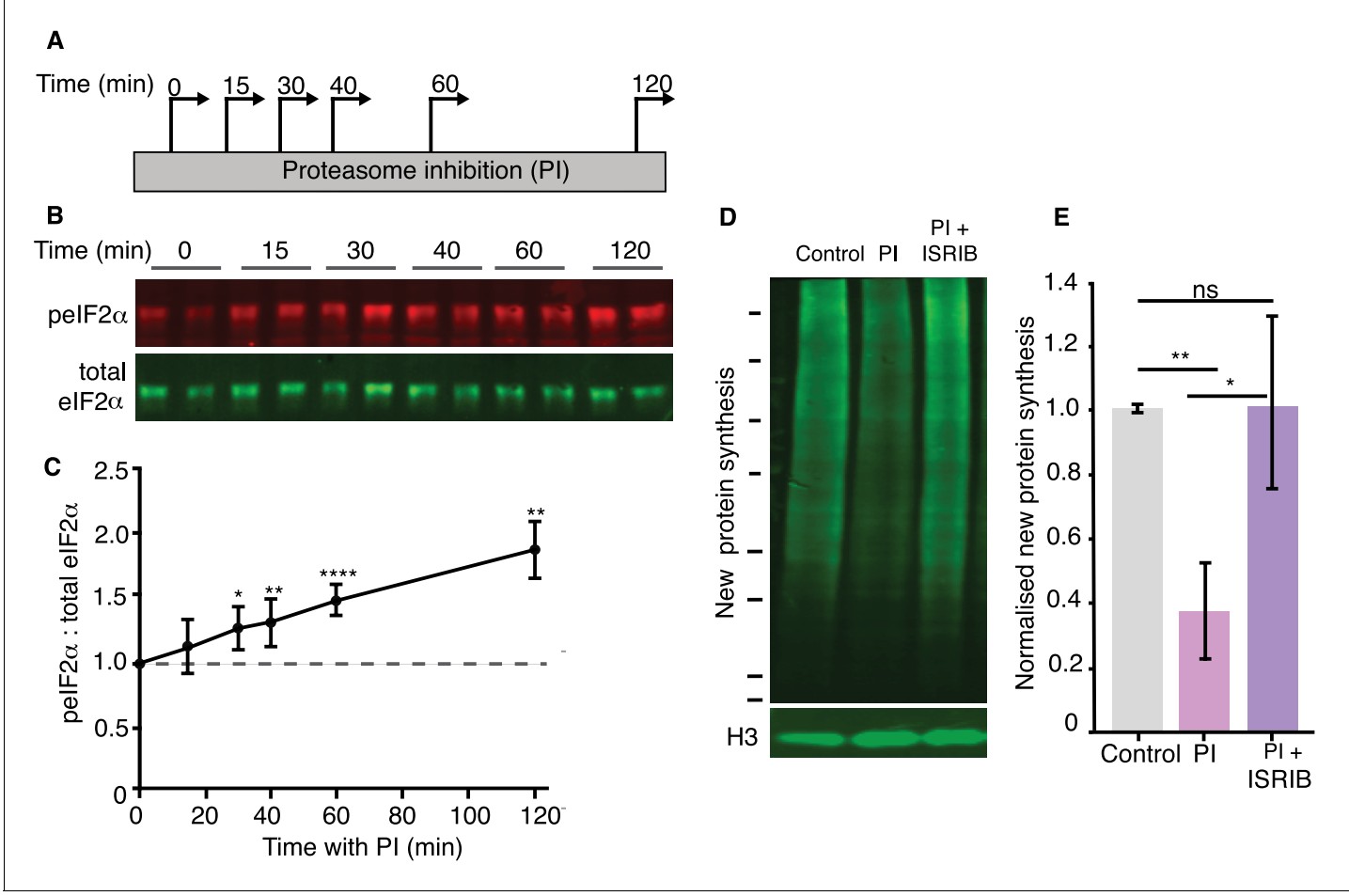

**Figure 2.** Reduced protein synthesis is mediated by eIF2α phosphorylation. (**A**) Scheme indicating experimental protocol: cultured hippocampal neurons were treated with a proteasome inhibitor (PI) and collected at the indicated time points. (**B**) Representative western blot showing the levels of phosphorylated (detected using a phospho-specific anti-eIF2α antibody) and total eIF2α under control conditions (t = 0) or following the indicated minutes of proteasome inhibition. (**C**) Analysis of experiments shown in B. PI treatment led to a significant increase in peIF2α levels. (unpaired t-test control vs each time point, Control vs 15 min p>0.05, control vs 30 min p<0.05, control vs 40 min p≤0.01, control vs 60 min p≤0.0001, control vs 120 min p≤0.01. (3 experiments). Error bars = SD. (**D**) Representative BONCAT western blot showing the metabolically labeled newly synthesized protein in green. Treatment with the small molecule inhibitor ISRIB rescued the PI-induced decrease in protein synthesis (2 hr of treatment). Molecular weight markers from top-to-bottom are 250, 150, 100, 75, 50, 37, 25, and 20 kD. Histone 3is shown as a loading control. (**E**) Analysis of experiment shown in D. The PI-induced protein synthesis inhibition (control vs. PI: p≤0.05) was rescued by ISRIB treatment (control vs. ISRIB, p>0.05, unpaired t-test), four experiments, Error bars = SD.

The online version of this article includes the following source data and figure supplement(s) for figure 2:

**Source data 1.** Source data for *Figure 2A-B, D*.
**Source data 2.** Source data for *Figure 2C, E*.
**Figure supplement 1.** Effect of transcription inhibition on protein synthesis after proteasome inhibition and phosphorylation of other translation factors.
**Figure supplement 1—source data 1.** Source data for *Figure 2—figure supplement 1B, C, F*.
**Figure supplement 1—source data 2.** Source data for *Figure 2—figure supplement 1A, D*.

(*Han et al., 2001*) we observed a dramatically reduced inhibition of protein synthesis induced by proteasome blockade with metabolic labeling detected by western blot (*Figure 3B,C*) or in situ labeling of cultured hippocampal neurons (*Figure 3D,E*). HRI deletion had no effect on the basal levels of protein synthesis in neurons or in brain tissue (*Figure 3E* and *Figure 3—figure supplement 1A,B*). The absence of HRI also significantly reduced the proteasome inhibition-induced increase in eIF2α phosphorylation (*Figure 3F* and *Figure 3—figure supplement 1C*), while the absence or inhibition of the other eIF2α kinases did not (*Figure 3—figure supplement 1D,E*). These data show that

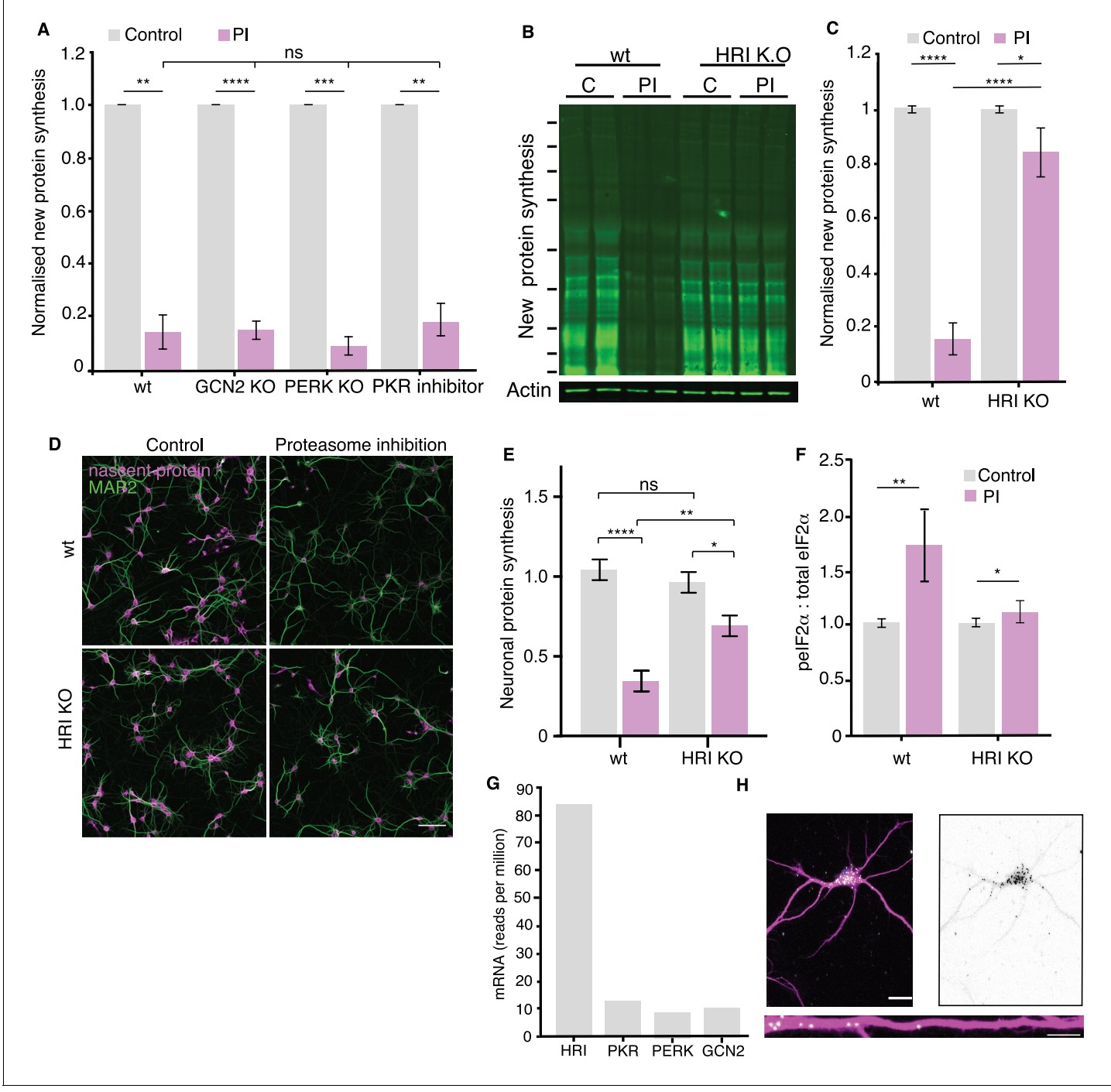

**Figure 3.** HRI kinase is responsible for the proteasome-inhibition induced increases in eIF2α phosphorylation. (**A**) Genetic deletion (KO) or inhibition of the eIF2α kinases GCN2, PERK or PKR did not rescue the inhibition of protein synthesis elicited by proteasome inhibition. (wt vs. kinase KO or inhibition: GCN2, unpaired t-test p≤0.0001, PERK, unpaired t-test p≤0.001 and PKR p≤0.01, respectively, n = 2 experiments, two biological replicates per KO). Error bars = SD. (**B**) Representative puromycilation western blot showing that genetic deletion of HRI kinase rescues the inhibition of protein synthesis. Molecular weight markers from top-to-bottom are 250, 150, 100, 75, 50, 37, 25, 20, and 15 kD. (**C**) Analysis of the experiment shown in B. The amount of PI-induced protein synthesis inhibition was significantly reduced in the HRI KO (unpaired t-test on wt PI vs KO. PI p≤0.0001, wt control vs. PI p≤0.0001, KO. control vs. PI p≤0.05, n = 5 experiments, error bars SD. (**D**) Metabolic labeling (magenta; using puromycilation, see Materials and methods) of the global nascent protein pool in cultured hippocampal neurons following proteasome inhibition showing that HRI KO neurons exhibit less PI-induced inhibition of protein synthesis. Scale bar = 100 µm. (**E**) Analysis of experiments like that shown in D. Protein synthesis in hippocampal neurons was significantly reduced in the HRI KO following proteasome inhibition (unpaired t-test on wt control vs. k.o. control p>0.05, wt

*Figure 3 continued on next page*

*Figure 3 continued*

PI vs k.o. PI p≤0.01, wt control vs. PI p≤0.0001, KO control vs. PI p≤0.05. n = 3 experiments, wt control n = 1104, wt-PI n = 955, ko control = 1457, ko-PI n = 1052 neurons). (F) Analysis of eIF2α phosphorylation in response to proteasome inhibition. PI treatment led to a significant increase in peIF2α levels. (unpaired t-test on wt control vs. PI: p≤0.01, t-test on HRI KO control vs. PI: p≤0.05. two experiments –see *Figure 3—figure supplement 1E*. (G) RNA-seq data from hippocampal slices indicate that HRI mRNA is the most abundant amongst the 4 eIF2α kinases (H) Representative fluorescence in situ hybridization image detecting HRI mRNA in neuronal somata and dendrites. Scale bar = 20 µm for somata and 10 µm for dendrites-See *Figure 3—figure supplement 2A–D*.

The online version of this article includes the following source data and figure supplement(s) for figure 3:

**Source data 1.** Source data for *Figure 3B*.
**Source data 2.** Source data for *Figure 3A, C, E, F*.
**Figure supplement 1.** Polysome and eIF2α phosphorylation data from eIF2α kinase knock-outs.
**Figure supplement 1—source data 1.** Source data for *Figure 3—figure supplement 1B,E*.
**Figure supplement 1—source data 2.** Source data for *Figure 3—figure supplement 1C,E*.
**Figure supplement 2.** HRI mRNA detection by fluorescence in situ hybridization.
**Figure supplement 2—source data 1.** Source data for *Figure 3—figure supplement 2B*.

a kinase, known primarily for its translational regulatory role in erythrocytes, plays a critical role in neuronal proteostasis.

## HRI protein is expressed at vanishingly low levels in neurons under control conditions

Given the surprising involvement of HRI, we next sought to measure the expression of HRI mRNA and protein in hippocampal neurons. Using data from a recent RNA-sequencing study (*Tushev et al., 2018*) we found that the HRI transcript is the most abundant of the 4 eIF2α kinases (*Figure 3G*) in hippocampal neurons. We used fluorescence in situ hybridization to validate and localize HRI mRNA in hippocampal neurons and tissue, and detected it in both somata and dendrites (*Figure 3H* and *Figure 3—figure supplement 2A–D*). The use of HRI knock-out tissue allowed us to identify in neurons, after immunoprecipitation, an HRI-specific band with one antibody (*Figure 4—figure supplement 1A*). We observed that the HRI expression level in brain was very low compared to the HRI levels detected in heme-rich tissues like liver or blood (*Figure 4—figure supplement 1B, C*).

We next considered whether proteasome inhibition might boost HRI expression by altering the level of HRI mRNA or protein. Quantification of HRI mRNA using droplet digital PCR (ddPCR) revealed no change in HRI mRNA levels following 2 hr of proteasome inhibition (*Figure 4A*), consistent with the transcription-independence of the protein synthesis reduction elicited by proteasome inhibition (*Figure 2—figure supplement 1A,B*). Following treatment with a proteasome inhibitor for 2 hr there was, however, a significant upregulation in neuronal HRI protein levels (*Figure 4B*, *Figure 4—figure supplement 1D*), suggesting that HRI might be a proteasome substrate. Although we were unable to detect appreciable polyubiquitylation of HRI (data not shown), using an in vitro assay with purified 20S proteasome and recombinant HRI we observed that HRI was degraded by the proteasome (*Figure 4—figure supplement 1E,F*).

The observation that HRI is a proteasome substrate suggests that proteasome inhibition stabilizes HRI protein levels. To estimate the magnitude of this effect, we metabolically pulse-labeled cultured neurons with $S^{35}$Methionine and immunoprecipitated HRI at different chase times under control and proteasome inhibition conditions. Surprisingly, we observed that HRI protein has an extremely short half-life under control conditions (~4.2 hr). In addition, the short-lived HRI protein was significantly stabilized by proteasome inhibition (*Figure 4C,D*). These data indicate that inhibition of the proteasome alone is sufficient to augment HRI protein levels and can account for at least part of the increase in HRI protein expression.

To explore the possibility that there is translational regulation of HRI, we placed both a fluorescent reporter (ZsGreen) and a HA-tagged HRI protein in a single doxycycline (DOX)-inducible bi-directional plasmid (*Figure 4E*). Transfection of the plasmid into neurons or HeLa cells followed by DOX treatment led to robust expression of ZsGreen in a large population of cells whereas the expression of HRI was absent or barely detectable in the same population (*Figure 4F,G* and *Figure 4—figure supplement 2A–C*). This result was unexpected, given that the use of a bi-directional

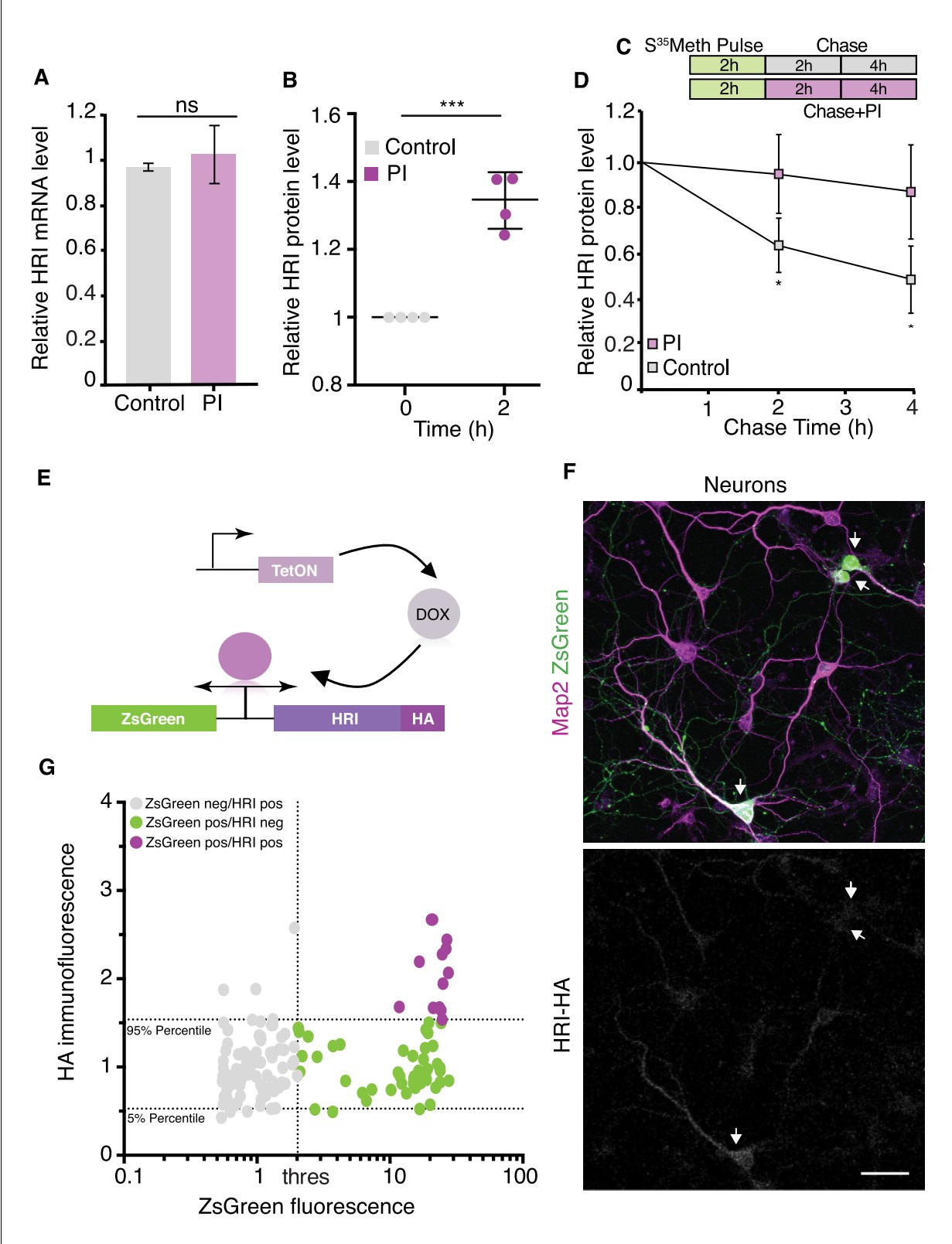

**Figure 4.** HRI exhibits low expression under basal conditions and increased expression following proteasome inhibition. (**A**) Analysis for ddPCR experiments showing there is no change in HRI mRNA level following proteasome inhibition, control vs. PI, (unpaired t-test, p>0.05 n=3, error bars = SD). Samples were normalized to ribosomal RNA. (**B**) Quantification of immunoprecipitated HRI protein levels under control conditions or after proteasome inhibition (PI). After PI (2 hr) a significant increase in HRI protein was detected in PI samples vs. control, experiments are normalized to the

*Figure 4 continued on next page*

*Figure 4 continued*

untreated condition (unpaired t-test p≤0.0001, n = 4 experiments, error bars = SD). (**C**) Scheme showing the experimental procedure, cultured neurons were labeled with $S^{35}$-Met for 2 hr, washed and collected after 2 and 4 hr + / - PI. (**D**) Analysis of immunoprecipitated and radiolabeled HRI. Under basal conditions the HRI half-life is ≈4 hr, its degradation is blocked by PI (unpaired t-test, 2 hr control vs PI p≤0.05, 4 hr control vs PI p≤0.05, four experiments, error bars = SD). (**E**) Scheme showing the doxycycline-inducible expression of the HA-tagged HRI protein. (**F**) Representative images of transfected neurons (arrowheads) with the bi-directional reporter resulting in robust expression of ZsGreen (green) but near absent expression of HRI (white) in the same population of cells. Also shown are somata and dendrites (labeled with an anti-MAP2 antibody). Scale bar = 50 µm. (**G**) Analysis of experiments in F, showing the correlation between ZsGreen fluorescence and HA (HRI) immunolabeling in individual neurons. Dotted horizontal lines indicate the area containing 90% of the HA immunofluorescence values of ZsGreen-negative (non-transfected) neurons (5–95% percentiles), the dotted vertical line indicates the threshold between ZsGreen-negative and ZsGreen positive cells. Grey dots represent ZsGreen-negative cells, magenta dots show neurons positive for ZsGreen but negative for HA, and green dots represent neurons positive for ZsGreen and above the HA 95% percentile of the ZsGreen-negative population (n = 149). See *Figure 4—figure supplement 2A*.

The online version of this article includes the following source data and figure supplement(s) for figure 4:

**Source data 1.** Source data for *Figure 4A, B, D, G*.
**Figure supplement 1.** Additional data on HRI expression following proteasome inhibition.
**Figure supplement 1—source data 1.** Source data for *Figure 4—figure supplement 1E*.
**Figure supplement 1—source data 2.** Source data for *Figure 4—figure supplement 1A, B, C, D, E, F*.
**Figure supplement 2.** Correlation of HRI expression and ZsGreen in primary neurons and HeLa cells.
**Figure supplement 2—source data 1.** Source data for *Figure 4—figure supplement 2A, C*.

promoter typically results in roughly equal populations of expressed proteins (*Vogl et al., 2018*). We next considered the possibility that the mismatch of ZsGreen and HRI-HA expression could be due to the compromised translation of HRI-HA. To address how HRI translation behaves relative to global translation, we used polysome profiling, comparing control and proteasome inhibitor-treated hippocampal cultured neurons. As shown in *Figure 5A* and consistent with the relative inhibition of translation observed in *Figure 1*, we observed a leftward shift in the translational profile from polysomes to monosomes following proteasome inhibition. We confirmed this shift using ddPCR to detect beta-actin mRNA levels in each fraction in control and PI conditions (*Figure 5B*). In contrast, HRI mRNA did not exhibit a leftward shift after proteasome inhibition, but rather exhibited a paradoxical rightward shift to the polysomal fractions (*Figure 5C*). An increase in polysome occupancy after proteasome inhibition has also been observed for a stress-related transcript (*Thomas and Johannes, 2007*; *Figure 5—figure supplement 1A*) which exhibit aberrant increases in translation, caused by a variety of manipulations, that are often mediated by uORFs or IRES sequences (*Di Prisco et al., 2014*; *Lang et al., 2002*). HRI, however, has a very short 5'UTR and no uORF or IRES is predicted (*Di Prisco et al., 2014*; *Mokrejs et al., 2010*; *Wu et al., 2009*).

An alternative mechanism we considered for the low basal levels of HRI protein is the presence of rare codons in the protein coding sequence (*Supplementary file 1*), which can result in reduced translational efficiency (*Cannarozzi et al., 2010*; *Chevance et al., 2014*; *Frumkin et al., 2018*; *Zhou et al., 2016*). We evaluated the rodent HRI sequence for the presence of rare codons and found a number of rare and extremely rare codons, HRI uses more rare codons than 86.78% of the genes expressed in the hippocampus (see Materials and methods), predicting compromised translation (*Figure 5D*). Indeed, when we expressed an HA-tagged HRI with optimized codon usage (HRI$_{opt}$-HA) we observed that the basal expression of HRI$_{opt}$-HA was higher (*Figure 5E,G* and *Figure 5—figure supplement 1B*) and HRI$_{opt}$-HA exhibited a reduced rightward shift to the polysome fraction following treatment with a proteasome inhibitor (*Figure 5—figure supplement 1C*). Taken together these data indicate that HRI protein expression is paradoxically enhanced under conditions in which global protein synthesis is reduced. Two complementary mechanisms act to enhance HRI levels; first, a stabilization of the protein via reduced degradation by the proteasome and, second, enhanced translation owing to the increased availability of rare codons.

## Stabilized HRI is constitutively active in neurons

In erythrocyte precursors, HRI kinase is activated by multiple mechanisms, including an increase in reactive oxygen species (ROS) or nitric oxide (NO) or reductions in cellular heme (*Chen, 2000*; *Chen and London, 1995*; *Chen et al., 1991*; *Igarashi et al., 2004*; *Uma et al., 2001*). Increases in ROS have been reported following proteasome inhibition (*Ding et al., 2006*). We thus tested

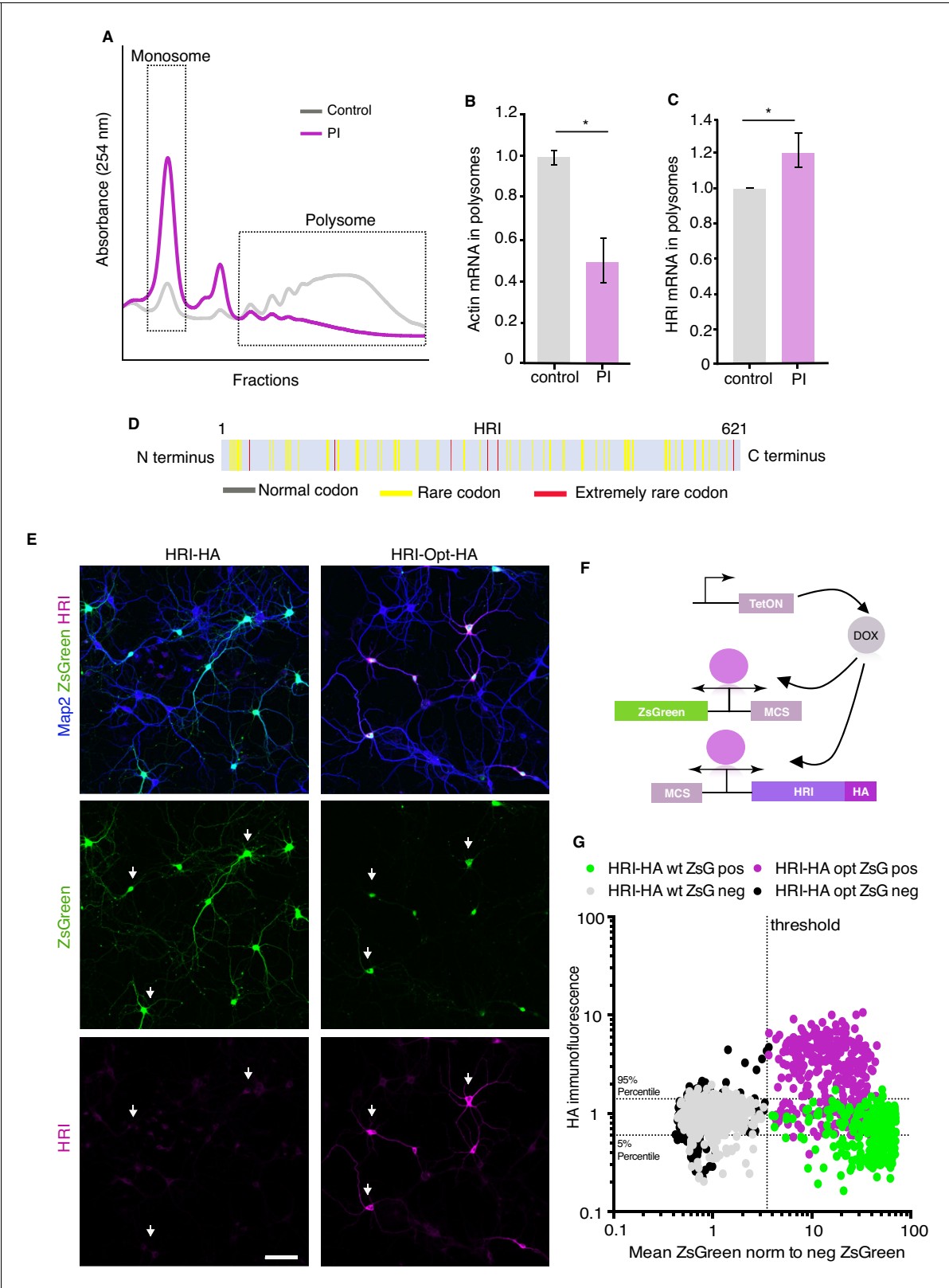

**Figure 5.** HRI exhibits a codon-dependent paradoxical shift to enhanced translation following proteasome inhibition. (**A**) Representative polysome profile showing the effect of proteasome inhibition on translation. PI led to a dramatic shift to the monosome fraction, reflecting reduced global translation. (**B**) Quantification of ddPCR experiments examining the abundance of actin mRNA in monosomes and polysomes, normalized to control levels. Following PI, actin mRNA exhibited the typical shift to the monosome fraction (unpaired t-test, p≤0.05, n = 2 experiments (three technical

*Figure 5 continued on next page*

*Figure 5 continued*

replicates per experiment). (**C**) Quantification of ddPCR experiments examining the abundance of HRI mRNA in monosomes and polysomes, normalized to control levels. Unlike the global RNA population, HRI mRNA exhibited a significant shift to the polysome fraction following PI, (unpaired t-test, $p \leq 0.05$, n = 2 experiments three technical replicates). Error bars = SD. (**D**) Scheme of the HRI protein showing an abundance of many rare and extremely rare codons, consistent with HRI's extremely low level of translation under basal conditions in neurons. HRI uses more rare codons than the 86.78% of the genes expressed in the brain (see Materials and methods). (**E**) Representative images showing the increased expression of codon-optimized HRI-HA in comparison with HRI-wt (Scale bar = 100 $\mu$m). (**F**) Scheme showing the plasmids used for the transfection shown in E, the plasmid shown in the *Figure 4E* was split in two parts, one expressing ZsGreen and another one expressing HRI, the plasmids were transfected in 1:5 ratio (ZsGreen:HRI) (**G**) Analysis of the experiments in E, showing the correlation between ZsGreen fluorescence and HA immunolabeling in individual neurons from dishes co-transfected with ZsGreen and HRI-HA either as wt sequence (HRI-HA) or codon optimized (HRIopt-HA). Dotted horizontal lines indicate the area containing 90% of the HA immunofluorescence values of ZsGreen-negative (non-transfected) neurons of the HRI-HA transfections (5–95% percentiles), the dotted vertical line indicates the threshold between ZsGreen-negative and ZsGreen positive cells. Grey and black dots are represent ZsGreen-negative (non-transfected) neurons from dishes co-transfected with HRI-HA and HRI-opt-HA, respectively. ZsGreen-positive neurons from dishes co-transfected with HRI-HA are shown in green and from dishes co-transfected with HRIopt-HA are represented in magenta. Neurons positive for ZsGreen and HA are located in the upper right quadrant (n = 2484 neurons).

The online version of this article includes the following source data and figure supplement(s) for figure 5:

**Source data 1.** Source data for *Figure 5B, C, G*.
**Figure supplement 1.** Additional data on HRI's codon-dependent paradoxical shift to enhanced translation following proteasome inhibition.
**Figure supplement 1—source data 1.** Source data for *Figure 5—figure supplement 1A, B, C*.
**Figure supplement 2.** Experiments addressing the mechanism of HRI activation.
**Figure supplement 2—source data 1.** Source data for *Figure 5—figure supplement 2A, B, C, D, E, F, G, H, L*.
**Figure supplement 2—source data 2.** Source data for *Figure 5—figure supplement 2I,K*.

whether the HRI activation that follows proteasome inhibition is due to increases in either ROS or NO by quenching ROS or blocking NO synthesis during MG132 treatment. We found that neither ROS scavengers nor NO synthase inhibitors prevented the proteasome inhibitor-induced decreases in protein synthesis (*Figure 5—figure supplement 2A–E*). We then examined whether heme metabolism drives the activation of HRI. The major enzymatic path that promotes the breakdown of heme are the heme oxygenases (HO1 and 2). Treatment with two different heme oxygenase inhibitors, however, also failed to affect the protein synthesis reduction elicited by proteasome inhibition (*Figure 5—figure supplement 2F,G*). These data suggest that enhanced heme breakdown, per se, is not responsible for HRI activation.

In erythryocytes, HRI is activated by reductions in cellular heme levels. We next considered the possibility that basal heme levels might be low in neurons, relative to erythrocytes, which could lead to constitutive kinase activity. We directly measured free heme in neurons and blood and found that while blood had predictably high levels of free heme, heme could not even be detected in cultured hippocampal neurons, and total hippocampal lysates exhibited heme levels orders of magnitude lower than blood (hippocampus; $36 \pm 8$ fmol/$\mu$g, blood; $126 \pm 18$ pmol/$\mu$g). We next tested if the measured amount of heme in blood could indeed inhibit HRI kinase activity in an in vitro assay using recombinant HRI and its substrate eIF2$\alpha$, together with $^{32}$P-ATP. We found that the addition of hemin (heme complexed with $Fe^{3+}$) led to a significant reduction in eIF2$\alpha$ phosphorylation by HRI (*Figure 6A*). Given the clear heme-induced inhibition of HRI, we directly tested whether the cytoplasmic regulatory context for HRI is different in neurons by adding recombinant HRI and its substrate eIF2$\alpha$, together with $^{32}$P-ATP to cellular lysates prepared from neurons or blood. We evaluated the phosphorylation of eIF2$\alpha$ and found that it was elevated in hippocampus when compared to blood, indicating that HRI is more active under basal conditions in the neuronal cytoplasmic context (*Figure 6B*). If the elevated HRI protein is constitutively active due to low neuronal heme levels, then the inhibition of protein synthesis induced by proteasome blockade should be amenable to rescue by exogenous heme in neurons. Indeed, we found that addition of hemin to cultured neurons resulted in a rescue of the protein synthesis inhibition (*Figure 6C*, *Figure 5—figure supplement 2H*), and a reduction in the phosphorylation levels of eIF2$\alpha$ (*Figure 5—figure supplement 2I,J*) caused by proteasome blockade. The addition of Hemin to HRI k.o. primary neurons had no effect on basal protein synthesis, ruling out an off-target effect of hemin (*Figure 5—figure supplement 2K,L*). Taken together, these data indicate that relatively low cytoplasmic heme levels in neurons favor the constitutive activity of HRI.

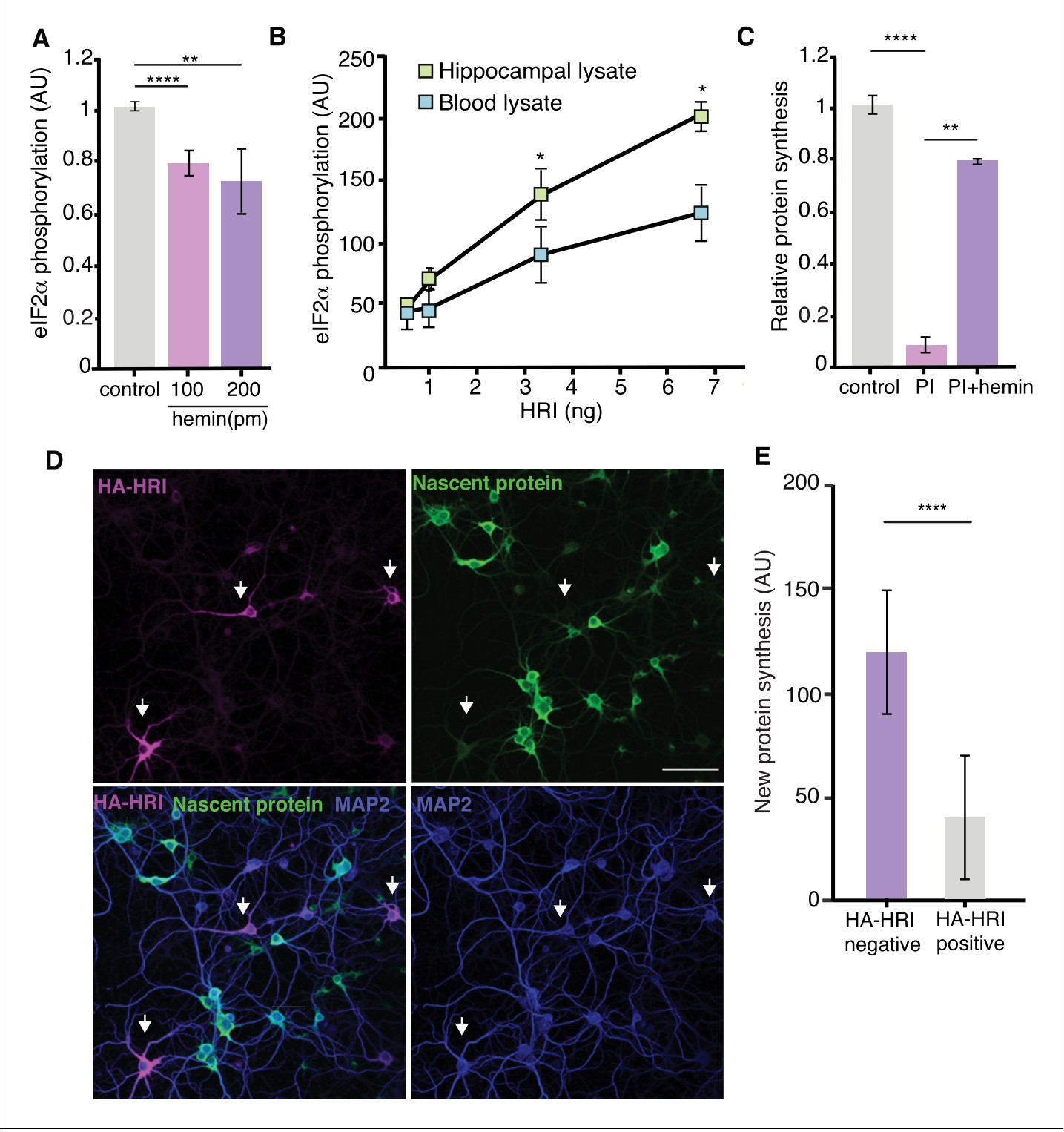

**Figure 6.** HRI activity is constitutive owing to a low heme content in neurons. (**A**) In vitro eIF2α phosphorylation assay by HRI; 100 and 200 pm of hemin were added to the reaction mixture (500 ng eIF2α and 3,3 ng of HRI), inhibiting the activity of the kinase (unpaired t-test control vs hemin 100pm p≤0.0001, control vs hemin 200pm p≤0.01). (**B**) In vitro eIF2α phosphorylation assay by HRI; a constant amount of eIF2α was added for each reaction combined with increasing amounts of HRI as indicated (See Materials and methods). Increasing amounts of HRI protein lead to significantly higher levels of eIF2α phosphorylation in hippocampal lysates, compared to blood lysates (the lysates were cleared of protein by heating 10 min at 60°C) (unpaired t-test 0.05 ng HRI Blood vs hippocampus p≥0.05, 0.1 ng HRI Blood vs hippocampus p≥0.05, 0.33 ng HRI Blood vs hippocampus p≤0.05, 66 ng HRI Blood vs hippocampus p≤0.05, n = 3 experiments). (**C**) Western blot analysis of experiments testing whether hemin (heme complexed with $Fe^{3+}$)

*Figure 6 continued on next page*

*Figure 6 continued*

can mitigate the effects of PI. Hemin significantly reduced the PI-induced protein synthesis inhibition. (PI vs. control, p≤0.0001; PI vs. PI +hemin, p≤0.01 Error bars = SD, three experiments) (D) Representative images showing that HA-HRIexpressing neurons (arrow) (green; anti-HA labeling) exhibit lower levels of protein synthesis (magenta) when compared to neighboring neurons, which do not express HA-HRI. Also shown are somata and dendrites (labeled with an anti-MAP2 antibody) and nuclei (labeled with DAPI). Scale bar = 50 µm. (E) Quantification of somatic protein synthesis levels in neurons expressing or lacking HA-HRI; HA-HRI positive neurons exhibited significantly lower levels of protein synthesis, (p≤0.0001, HA negative neurons n = 852 and HA positive neurons n = 253, unpaired t-test).

The online version of this article includes the following source data and figure supplement(s) for figure 6:

**Source data 1.** Source data for *Figure 6A, B, C, E*.
**Figure supplement 1.** Analysis of the data shown in *Figure 6*.

The observation that the neuronal cytoplasmic context favors HRI activity suggests that mere expression of HRI could serve as a regulatory switch for protein synthesis. We revisited the HRI over-expression experiments to examine protein synthesis in neurons expressing HRI or not. Brief meta-bolic labeling of transfected neurons to measure protein synthesis indicated that the small neuronal population that expressed appreciable levels of HRI exhibited significantly reduced protein synthe-sis, on average, relative to other cells that did not express HRI (*Figure 6D,E*). Furthermore, small increases in HRI expression can have a significant impact on protein synthesis (*Figure 6—figure supplement 1*). This indicates that, in neurons, the cytoplasmic environment is permissive for the consti-tutive activity of HRI upon expression. In addition, HRI's potent effect on global translation provides an explanation for why its expression is usually maintained at low levels in neurons. These features of HRI, its basal very low expression, its short half-life and the boost in protein levels upon proteasome inhibition, thus make it both an optimal sensor and effector for proteostasis during situations of compromised proteasome activity.

## Discussion

Here we report that after brief period of proteasomal inhibition (via a pharmacological or genetic manipulation), neuronal protein synthesis is coordinately reduced via the actions of the eIF2α kinase 1, or heme-regulated inhibitor kinase. Taken together, our data indicate the following sequence of events unfolds to restore proteostasis: i) proteasome inhibition (PI) stabilizes the extremely short-lived neuronal HRI protein, ii) endogenous low levels of neuronal heme result in constitutive activa-tion of HRI which increases the phosphorylation of eIF2α and reduces translation initiation, iii) the ini-tial decrease in general translation may lead to an increase in the availability of charged rare tRNAs (*Saikia et al., 2016*), further increasing the translation of HRI mRNA and ensuring ongoing phos-phorylation of eIF2α and as a result, reduced global translation.

The above-described mechanism allows for the coordination of protein synthesis and degradation in neurons, so that global protein levels stay balanced. We found that the same feedback mechanism can be detected in isolated neuronal processes, including dendrites and potentially axons. Local syn-thesis and degradation of proteins occurs in dendrites and axons via localized mRNAs, ribosomes and components of the UPS (e.g. [*Bingol and Schuman, 2006*; *Campbell and Holt, 2001*; *Hafner et al., 2019*; *Holt and Schuman, 2013*]). It is not known, however, the spatial scale over which this proteostasis operates in axonal or dendritic compartments. In addition, a completely open question in all cells is the how proteostasis is accomplished at the level of individual proteins-of-interest.

Our data indicate that the above mechanisms cleverly cooperate to enhance the expression and activity of HRI in response to reduced proteasome activity. These mechanisms make HRI an opti-mized sensor and effector for sensing and responding to proteasome inhibition. In this regard, it is interesting to note that while the HRI transcript is detected at higher levels than the other 3 eIF2α kinases in hippocampal neurons, the HRI protein is expressed at very low levels under basal condi-tions, suggesting strong cellular pressure to keep HRI protein expression low. We observed, though, that HRI levels exhibit an increase upon proteasome inhibition, consistent with our observation that HRI can be degraded by the proteasome in vivo and in vitro. In addition, attempts to over-express HRI via transfection were largely thwarted. We evaluated the transcript for the presence of rare codons which can reduce translational efficiency (e.g. [*Frumkin et al., 2018*]). Indeed, HRI mRNA

possesses many rare and extremely rare codons; expression of a codon-optimized HRI resulted in higher levels of basal translation. In addition, in polysome profiling experiments, we found that HRI mRNA exhibited a paradoxical shift to the polysome fraction following proteasomal inhibition, while the bulk of cellular mRNAs exhibited a shift to the monosome fraction, consistent with their reduced global translation. Owing to the apparent absence of uORFs in the 5'UTR HRI, we propose that rare codon availability could explain the atypical behavior of HRI mRNA following PI (*Saikia et al., 2016*), in support of this idea, the codon-optimized HRI failed to exhibit the shift to the polysome fraction upon PI.

In erythrocytes, HRI is activated by reduced heme levels to coordinately reduce the synthesis of α- and β-globin (*Han et al., 2001*); together α-globin, β-globin, and heme, assembled in a 2:2:4 ratio, make up hemoglobin. Getting the stoichiometry just right is important for red blood cells (RBCs) since an overabundance of any single component is cytotoxic to RBCs and their precursors. We found that heme levels are extremely low or undetectable in neurons, indicating that once expressed at appreciable levels, HRI will be constitutively active. Indeed, we observed significantly elevated HRI activity in a control hippocampal lysate when compared to a control blood lysate. Importantly and definitively, we observed that the PI-induced protein synthesis inhibition was rescued by the addition of hemin.

Taken together, multiple mechanisms act in concert to tightly control the dynamic range of HRI. Under control conditions HRI exhibits barely detectable levels of expression and activity due to its high turn-over and low translation rates. The inhibition of the proteasome is sensed by the stabilization of HRI and the low neuronal heme content means that HRI can respond immediately to the environment created by proteasome blockade. These data further suggest that physiological and pathophysiological conditions that give rise to proteasome dysfunction may favor HRI expression and activity leading to a decrease in protein synthesis.

The role of HRI as a primary mediator of the feedback between the degradation and protein synthesis machinery is surprising given the prominent role the other eIF2α kinases (GCN2, PERK and PKR) are known to play in neuronal translational regulation (*Cagnetta et al., 2019*; *Costa-Mattioli et al., 2005*; *Hughes and Mallucci, 2019*). Indeed, all four of the eIF2α kinases have been associated with the phosphorylation of eIF2α in response to proteasome inhibition. These studies were performed with cell lines of non-neuronal origin or mouse embryonic fibroblasts (MEF) (*Jiang and Wek, 2005*; *Yerlikaya et al., 2008*; *Zhang et al., 2010*). In addition, the activation of mTOR pathway following UPS inhibition has also been linked to a decrease in protein synthesis (*Zhang et al., 2014*). In our experiments, HRI is clearly the major kinase responsible for the proteasome inhibition-induced feedback regulation of neuronal protein synthesis; we think it is likely, however, that the other eIF2α kinases also play a role, potentially when HRI is absent or during periods of prolonged proteasome inhibition. Parsing the unique and complementary roles of these kinases will have to take into account the unique translation/degradation rates of each cell type together with the specific expression the eIF2α kinases, protein chaperones and heme content (*Uma et al., 1999*).

We note that HRI's role in neuronal function is poorly explored, save for two reports of HRI influencing the translation of synaptic proteins (*Ramos-Fernández et al., 2016*) and influencing learning (*ILL-Raga et al., 2013*). In addition, the regulation by heme highlights an important element to consider in the context of neuronal function, given emerging data that point to dysregulation of iron homeostasis in disease (*Salvador, 2010*). Dysfunctional proteasome activity occupies a central position in many neurodegenerative diseases, and increased phosphorylation in eIF2α has been detected in postmortem brains of AD patients (*Chang et al., 2002*). Prion infection also inhibits proteasomal activity and increases polyubiquitin levels by 2–3 times (similar to the levels observed in our conditions of study- see *Figure 1D*; *Kang et al., 2004*). Strikingly, reperfusion after ischemic stroke leads to a profound proteasome inhibition in brain tissue (*Hochrainer et al., 2012*). In addition, in hemorrhagic strokes heme is liberated in the brain (*Babu et al., 2012*; *Prabhakaran and Naidech, 2012*) and causes neuronal toxicity. The expression of HRI in neurons and its susceptibility to heme regulation should thus be considered in the realms of neurodegeneration and iron dyshomeostasis.

# Materials and methods

## Key resources table

| Reagent type (species) or resource | Designation | Source or reference | Identifiers | Additional information |
|---|---|---|---|---|
| Gene (Rat or Mouse) | -HRI<br>-HRI-opt<br>-ZsGreen | GeneScript<br>Clone ID: OMu18622C | Entrez Gene: NM_013557.2 | |
| Strain, strain background (*Escherichia coli*) | BL21 (DE3) | Sigma-Aldrich | CMC0016 | Electrocompetent cells |
| Genetic reagent (Rat or Mouse) | -Primary Neurons | Other | | Animals obtained from |
| Cell line (Human) | -HeLa (RRID:CVCL_0030) | -ATCC | | |
| Transfected construct (mouse, Rat) | -HA-HRI<br>-HRI-HA<br>-HRI-HA-opt<br>-ZsGreen<br>-ZsGreen-HRI-HA<br>-Ubiquitin K48R | This paper | | Constructs used to over<br>HRI or ubiquitin genes i<br>primary neurons.<br>The constructs of HRI w<br>different tags can be ob<br>from Erin Schuman's<br>Laboratory (MPI-BR) |
| Biological sample (Mouse) | -Blood (mouse wt and HRI K.O)<br>-Liver (mouse wt and HRI K.O)<br>-Brain (mouse wt and HRI K.O) | DOI:<br>10.1093/emboj/20.23.6909 | | |
| Antibody | -Rabbit polyclonal anti-peIF2α | Invitrogen | 44728G<br>RRID:AB_1500038 | (1:1000) WB |
| Antibody | -Mouse monoclonal eIF2α | Cell signalling | 2103 | (1:1000) WB |
| Antibody | -Rabbit polyclonal peIF4B | Cell signalling | 5399 | (1:2000) WB |
| Antibody | -Rabbit Monoclonal p4EBP1 | Cell signalling | 2855<br>RRID:AB_560835 | (1:1000) WB |
| Antibody | -Rabbit polyclonal peIF4G | Cell signalling | 2441 | (1:25000) WB |
| Antibody | -Rabbit polyclonal HRI | Millipore | 07–728<br>RRID:AB_441964 | (1:1000) WB |
| Aantibody | -Rabbit polyclonal anti-actin | abcam | Ab8227<br>RRID:AB_2305186 | (1:5000) WB |
| Antibody | -Rabbit polyclonal anti-H3 | Abcam | AB18521<br>RRID:AB_732917 | (1:10000) WB |
| Antibody | -Rabbit polyclonal anti-biotin | SIGMA | 31852 | (1:10000) WB |
| Antibody | -Mouse monoclonal anti-puromycin | Kerafast | 3RH11 | (WB, 1:1000, IF 1:3500) |
| Antibody | -Guinea pig polyclonal anti-MAP2 | Synaptic Systems | 188004 | (IF 1:1000) |
| Antibody | -Rabbit polyclonal anti- HA-tag | Rockland | 600-401-384 | (IF 1:2000) |
| Antibody | -Goat polyclonal anti-mouse or anti-rabbit IR680 or IR800 | Licor | -Goat anti-mouse or anti-rabbit IR680 or IR800 | (WB 1:5.000,) |
| Antibody | -Goat polyclonal anti-guinea pig Dylight405 | Dianova | 106-475-008<br>RRID:AB_2337434 | (IF 1:1000) |
| Antibody | -Goat polyclonal anti-guinea pig-Alexa488 | Dianova | 106-546-003<br>RRID:AB_2337441 | (IF 1:1000) |
| Antibody | -Alexa488, polyclonal goat anti-rabbit Alexa647 or -Alexa546 | ThermoFisher | R37116, A32733, A-21085 | (IF all 1:1000) |
| Peptide, recombinant protein | -eIF2a<br>-HRI | Abcam | ab95932<br>ab131665 | |
| Peptide, recombinant protein | —20S proteasome | Enzo | BML-PW8720-0050 | |

*Continued on next page*

*Continued*

| Reagent type (species) or resource | Designation | Source or reference | Identifiers | Additional information |
|---|---|---|---|---|
| Commercial assay or kit | -Free heme measurement kit | Sigma | MAK316 | |
| Chemical compound, drug | -Puromycin<br>-Doxycycline (Doxo)<br>-MG132<br>-Lactacystin<br>—7-Nitroindazole (7-NA)<br>-S-methyl-L-thiocitrulline (L-SMTC)<br>-N-acetyl-cysteine (NAC)<br>-Ascorbic Acid (AA)<br>-MitoTempo<br>-Trolox<br>-Actinomycin-D (ActD)<br>-FePPIX (hemin)<br>-CoPPIX | Sigma | -Puromycin (P8833)<br>-Doxycycline (D9891)<br>-MG132 (M8699)<br>-Lactacystin (L6785)<br>—7-Nitroindazole (N7778)<br>-S-methyl-L-thiocitrulline (M5171)<br>-N-acetyl-cysteine (A9165)<br>-Ascorbic Acid (A1968)<br>-MitoTempo (SML0737)<br>-Trolox (648471)<br>-Actinomycin-D (A1410)<br>-FePPIX (hemin) (H9039)<br>-CoPPIX (C1900) | |
| Chemical compound, drug | -SnMPIX<br>-ZnPPIX<br>-NG mono methyl L-Arginine (L-NMMA)<br>-The PKR inhibitor | Cayman | -SnMPIX (19071)<br>-ZnPPIX (14483)<br>-L-NMMA (10005031)<br>-The PKR inhibitor (15323) | |
| Genetic reagent (*Mus musculus*) | -PERK K.O.<br>-GCN2 K.O. | Jackson | -PERK K.O. (line #009340) RRID:IMSR_JAX:009340<br>-GCN2 K.O. (line #008240) RRID:IMSR_JAX:008240 | |
| Genetic reagent (*Mus musculus*) | -HRI K.O.<br>Obtained from the laboratory of Dr, Jane Chen (Harvard-MIT) | PMID:11726526 (*Han et al., 2001*) | | |
| Software, algorithm | https://github.molgen.mpg.de/MPIBR-Bioinformatics/CodonUsage | https://github.molgen.mpg.de/MPIBR-Bioinformatics/CodonUsage | | The kinase gene list was generated by a keyword search in gene descripti Analysis and plotting scripts can be found in this repository |
| Sequence-based reagent | HRI Taqman Assay | IDT | Rn.PT.58.5930825 | Used for ddPCR |
| Sequence-based reagent | Rn18s rRNA 18 s Taqman Assay | Thermo-fisher | Mm0427757_s1 | Used for ddPCR |
| Sequence-based reagent | Actin beta Taqman Assay | IDT | Rn.PT.39a.22214838.g | Used for ddPCR |
| Sequence-based reagent | Atf4 Taqman Assay | IDT | Rn.PT.58.13690870.g | Used for ddPCR |
| Sequence-based reagent | HRI-HA | IDT | PCR primer one for HRI-hemagglutinin tag | Used for ddPCR CAGCTACTGCAGAGC |
| Sequence-based reagent | HRI-HA | IDT | PCR primer two for HRI-hemagglutinin tag | Used for ddPCR GCGTAATCTGGAACA |
| Sequence-based reagent | HRI-HA | IDT | Probe | Used for ddPCR /56-FAM/AGCCTCCTT/ TCGCAGGACAAAGGG |
| Sequence-based reagent | HRI-HA-opt | IDT | PCR primer 1 HRI-hemagglutinin tag optimized | Used for ddPCR TCCAGAGCGAGCTGT |
| Sequence-based reagent | HRI-HA-opt | IDT | PCR primer 2 HRI-hemagglutinin tag optimized | Used for ddPCR GCATAGTCAGGCACA |
| Sequence-based reagent | HRI-HA-opt | IDT | Probe HRI-hemagglutinin tag optimized | Used for ddPCR /56-FAM/ACCACCGG AACGTGAACCTGACC |

## Primary neuronal cultures

Neuronal cultures were prepared and maintained as described in *Aakalu et al. (2001)*. Briefly, the cortex from E17 mice or hippocampus from P1 rats were dissected, dissociated with papain (Sigma) and plated on poly-D-lysine coated dishes. Neurons were maintained at 37°C and 5% $CO_2$ in Neurobasal-A plus B27 and GlutaMAX, (Life Technologies). One day after plating neurons were incubated with 3 µM AraC (Sigma) for two days, then the AraC was removed by changing the media. Unless otherwise indicated, neurons were used for experiments at DIV 14–21. For the experiments with KO mice cortical neurons were used, hippocampal neurons where used for the rest of the experiments.

## Transgenic animals

PERK KO (line #009340) and GCN2 KO (line #008240) mice were purchased from the Jackson laboratory. The HRI KO mouse line was kindly provided by Dr. Jane-Jane Chen (*Han et al., 2001*). All lines were maintained as heterozygotes. For neuron cultures, E17 pups from KO/WT x KO/WT breedings were genotyped as indicated by the manufacturer or (*Han et al., 2001*). Cortices from WT and KO animals were pooled separately and dissociated and plated as described above. Procedures involving animals were performed according to German and Max Planck Society animal care guidelines and approved by and reported to the local governmental supervising authorities (Regierungspräsidium Darmstadt).

## Metabolic labeling, BONCAT and tissue preparation

Neurons were grown and treated with various agents (proteasome inhibitors and other inhibitors) in Neurobasal A (Invitrogen). After treatment, the medium was changed to Neurobasal A lacking Methionine and supplemented with 4 mM AHA + / - treatments for 10 min. Neurons were washed 3x with PBS and lysed in PBS with 1% (w/v) Triton X100, 0.4% (w/v) SDS, protease inhibitors w/o EDTA (Calbiochem, 1:750) and benzonase (Sigma, 1:1000), heated at 95°C and centrifuged. BONCAT was performed as described in *Dieterich et al. (2006)*. In brief, 30 µg of total protein was subjected to a click reaction with 300 µM Triazol (Sigma, ref 678937), 50 µM biotin-alkyne tag (Thermo, ref B10185) and 83 µg/mL CuBr at 4°C o.n. in the dark. Biotinylated proteins were detected by Immunoblot. For puromycylation, puromycin was added to the culture media + / - treatments to a final concentration of 1.5 ng/µl for 10 min, then the neurons were washed 2x with PBS and lysed. For phosphorylation studies a phosphatase inhibitor cocktail (Invitrogen) was added to the lysis buffer. Tissue from HRI KO or wild-type mice was homogenized in lysis buffer (1% (w/v) Triton X100, 1% (w/v) SDS, protease inhibitors (1:750) and benzonase (1:500), heated to 95°C and centrifuged. SDS-PAGE was performed with 3 µg of lysate and proteins were subsequently blotted onto PVDF membranes.

## Metabolic labeling of dendrites and axons (neurites)

1 million dissociated rat hippocampal neurons were plated into 6-well filter inserts with a 3 µm pore size (Falcon). The day after plating AraC (3 µM; Sigma) was added to the cultures and medium was changed two days later. At DIV8 the proteasome inhibitor MG132 (10 µM) was added to the medium. After 2 hr of treatment the upper part of the filter was carefully scraped with PBS to remove the cell bodies. The dendrites and axons remaining on the other side of the filters were washed 3 times in Neurobasal A-lacking Methionine and incubated for 5 min in Neurobasal A-lacking Methionine supplemented with 4 mM AHA and MG132 (10 µM). Dendrites were collected in lysis buffer and BONCAT was performed as described above. For an example of the quality of compartment separation by the membranes see Figure S1 in *Biever et al. (2020)*.

## Cell viability

Neurons were plated in 96 mw plates (20.000 neurons per well) and grown in Neurobasal A without phenol red. The viability was measured with a cell counting kit-8 (Sigma) following the manufacturer instructions. Cell counting solution was added to the media for 1 hr after 2 hr of PI incubation (3 hr of PI incubation in total).

## In situ puromycylation, RNA-FISH and immunocytochemistry

In puromycylation experiments, rat hippocampal or mouse cortical neuron cultures grown on MatTek glass bottom dishes were treated ±proteasome inhibitors and other drugs for the indicated times before puromycin was added at a final concentration of 3.0 µM for 5 min then cells were washed 2 times with PBS-MC before fixation and immunocytochemistry. In FISH or immunocytochemistry experiments without puromycylation the incubation medium was removed and cells were immediately fixed for 20 min with PBS containing 4%Sucrose and 4%PFA, pH = 7.4. Panomics RNA-FISH was performed as described previously (*Cajigas et al., 2012*) using the ViewRNA ISH Cell Assay Kit with 550 dye detection (ThermoFisher). For immunocytochemistry, cells were permeabilized for 15 min with 0.5% Triton in blocking buffer (BB) (PBS with 4% goat serum), blocked in BB for 1 hr and incubated with primary antibodies in BB for 1 hr. After washing, secondary antibodies in BB were applied for 30 min followed, when necessary, by a 3 min incubation with 1 µg/µl DAPI in PBS to stain nuclei. Cells were washed and imaged in PBS and mounted with Aquapolymount (Polysciences) for storage.

## Proteasome peptidase activity

Proteasome activity was measured as described (*Martìn-Clemente et al., 2004*). In brief, after treatment with proteasome inhibitors the neurons were washed 2x with cold PBS and lysed (50 mM Hepes pH 7.4, 50 mM NaCl, 5 mM EDTA, 10 µM leupeptin, 1 µg/ml pepstatin and 1 mM PMSF). Lysates were homogenized and centrifuged at 3000 g for 10 min at 4˚C. Supernatants were immediately used for measurement of proteasome peptidase activity by cleavage of the fluorogenic peptide *N*-Suc-LLVY-MCA (Sigma), enzymatic activity were acquired for 2 hr. Fluorescence was detected in a micro-plate reader (TECAN) with 400/505 filters. To estimate background signal produced from cleavage by other proteases, proteasome inhibitors were added to lysates from untreated controls.

## Antibodies

The following antibodies were used for immunofluorescence (IF) and/or immunoblotting (IB) at the indicated dilutions: Rabbit anti-peIF2α (for PLA 1:6000 #44728G Invitrogen, and IB, 1:1000, Cell Signaling), mouse anti-eIF2α (IB, 1:1000, Cell Signaling). peIF4B, p4EBP1, peIF4G (IB:1:2500, cell signaling). Most of the commercially available antibodies that should recognize rodent HRI failed to robustly detect the protein in neurons. Nevertheless, the use of HRI knock-out tissue allowed us to identify in neurons, after immunoprecipitation, an HRI-specific band only with one HRI antibody: 07–728, Millipore. Rabbit anti-actin (1:5000, Abcam), rabbit anti-H3 (1:10000, Abcam) rabbit anti-biotin (IB, 1:1000), mouse anti-puromycin (IB, 1:1000, IF 1:3500, Kerafast,), guinea pig anti-MAP2 (IF 1:1000, Synaptic Systems), rabbit anti- HA-tag (IF 1:2000, Rockland) Goat anti-mouse or anti-rabbit IR680 or IR800 (IB, 1:5.000, Licor), goat anti-guinea pig Dylight405 (IF 1:1000, Dianova), goat anti-guinea pig-Alexa488, and goat anti mouse-Alexa546 or -Alexa488, goat anti-rabbit Alexa647 or -Alexa546 (IF all 1:1000, ThermoFisher).

## Immunoprecipitation

The anti-HRI antibody was conjugated with Dynabeads (protein G, Invitrogen) o.n. at 4˚C. 30 million mouse cortical neurons (DIV10) were used per IP reaction and lysed in 50 mM NaCl, 0.5% NP40 buffer. The lysate was centrifuged at 16,000 g for 15 min. The supernatant was incubated with the antibody-bound beads o.n. at 4˚C. After binding, the beads were washed 4x times for 20 min with lysis buffer at 4˚C. Bound proteins were eluted from the beads with 0.1M glycine pH = 2.5, the eluates neutralized, subjected to SDS-PAGE and transferred to PVDF membranes.

## DNA constructs

Rat HRI was cloned into the Tet on-off system from Clontech (Tet-On 3G Bidirectional Inducible Expression System with ZsGreen1) and an HA tag was added. For some experiments (as indicated in the text) ZsGreen sequences were removed from the plasmid (HRI-HA Tet-On minZsGreen) leaving only HRI and HA inserted (constructs synthesized by GenScript). Codon-optimized versions were made in HRI-HA Tet-On minZsGreen plasmids using GenScript algorithms including optimization of the HA tag. HA-Ubiquitin K48R-G76A (construct synthesized by GenScript).

## Transfections

Neurons were transfected with Magnetofectamine (Life Technologies) using 1.5 µg of DNA per 40K neurons for 1 hr in Neurobasal A (Invitrogen). After transfection, the transfection medium was replaced by conditioned medium. When the plasmid HRI-HA Tet-On minZsGreen was used (see constructs above), it was mixed in a 5:1 proportion with a plasmid containing only ZsGreen under the same bidirectional, inducible promoter to control for transfection efficiency (ten times excess of HRI-HA Tet-On minZsGreen). One week after transfection, Doxycycline (1 µg/1 µl, Invitrogen) was added to the cultures for 16 hr before puromycylation. HeLa cells (ATCC) were grown in Dulbecco's modified eagle's medium (DMEM, Invitrogen) with 10% fetal calf serum (FCS) at 37°C in an atmosphere with 5% $CO_2$ and transfected with lipofectamine (Invitrogen) according to the manufacturer's instructions. Contamination with mycoplasma was evaluated by PCR (eMyco kit, Intron Biotechnology). After 24 hr of transfection Doxycycline was added and experiments were performed.

## HRI codon usage

A Codon adaptation index (CAI) (*Sharp and Li, 1987*) was used to calculate the relative abundance of rare codons in HRI compared with both the transcripts expressed in the hippocampus (12643) and 566 kinases expressed in the hippocampus (*Newman et al., 2016*). HRI has a lower CAI value than the median for all genes (median for EIF2αk1 = 0.7637 vs. all genes = 0.7995 or kinases = 0.8005) in both cases CAI value for EIF2αk1 is in the lowest quartile of the observed CAI distributions. Compared to the distributions EIF2αk1 has the following percentile: 1.) all genes => 13.2238 (EIF2αk1 uses more rare codons than 86.78% of the genes expressed in the hippocampus) 2.) all kinases => 9.5456 (EIF2αk1 uses more rare codons than 90.46% of the kinases expressed in the hippocampus). The mouse reference Transcriptome was downloaded from NCBI RefSeq repository (ftp://ftp.ncbi.nlm.nih.gov/refseq/M_musculus/mRNA_Prot/). Gene expression in mouse acute hippocampal slices in control conditions was determined by RNA sequencing experiment previously conducted in the lab (*You et al., 2015*). The kinase gene list was generated by a keyword search in gene description. Analysis and plotting scripts can be found in the following repository https://github.molgen.mpg.de/MPIBR-Bioinformatics/CodonUsage. (*Alvarez-Castelao, 2020*; copy archived at https://github.com/elifesciences-publications/MPIBR-Bioinformatics-CodonUsage).

## ddPCR

RNA was isolated from crude lysates or ribosomal fractions with QIAzol, following the manufacturer's instructions. 2–4 ng of RNA were treated with 4 units of DNAse for 20 min at 37°C and inactivated for 15 min at 75°C. The RNA was converted to cDNA using SuperScript IV (ThermoFisher) with a mixture of Random primers and dT15V primers. 1 µl out of 10 µl of the resulting cDNA was used for ddPCR (Bio-rad) with Taqman probes (Integrated DNA Technologies; see key Resources table for the sequences) following the manufacturer's instructions. The specificity of the primers for the HA-tagged versions of HRI was confirmed by ddPCRs in non-transfected neurons. The mRNA counts of each gene were normalized to the rRNA content of the corresponding fraction (ddPCR for rRNA was performed with cDNA diluted 1:10000). The normalized mRNA content of the fractions 7 to 11 (polysome) was related to the fractions 2–4 (monosome) with the following formula: (mRNA polysome)/(mRNA monosome + mRNA polysome) The disome was excluded from the calculations as its role in mRNA translation is not clear.

## In vitro degradation assay

200 ng of recombinant HRI (SignalChem) or 500 ng of recombinant eIF2α (abcam) were incubated with 2 µg of purified 20S proteasome (Enzo) in 20 mM Hepes pH 7.4/2 mM EDTA/1 mM EGTA for the indicated times at 37°C. When indicated, the 20S proteasome was incubated with 10uM MG132 for 30 min prior to its incubation with the kinase. Protein mixtures were loaded onto SDS-PAGE gels and stained o.n. with SYPRO Ruby Protein Gel Stain as indicated by the vendor. Gels were imaged using Gel Doc XR (Bio-Rad). In the figure the relative degradation of HRI+20S to HRI+20S+PI is shown.

## Pharmacological treatments

Proteasome inhibitors MG132 and Lactacystin were purchased from Sigma, dissolved in DMSO and used from 10 to 25 µM as indicated, unless otherwise indicated the inhibitor used in the experiments was MG132. Porphyrins were dissolved in DMSO and purchased from Millipore (CoPPIX), Sigma (FePPIX or hemin), or Cayman (SnMPIX and ZnPPIX), 4, 6 or 12 µM of each porphyrin was tested. In the figures always the 4 µM dosage is shown. Porphyrins were added at the same time as the pro-teasome inhibitors. NO inhibitors were added 1 hr before proteasome inhibitors and kept during the proteasome inhibitor treatment. L-NMMA (NG mono methyl L-Arginine, Cayman) was dissolved in water and used at 2 and 6 mM. 7-Nitroindazole (7-Ni, Sigma) was dissolved in DMSO and used at 100 and 200 µM. N-Omega-Nitro-L-Arginine (L-NNA, Sigma) was dissolved in water and used at 100 and 200 µM. S-methyl-L-thiocitrulline (L-SMTC, Sigma) was dissolved in water and used at 10 and 50 µM. ROS inhibitors were added 1 hr before proteasome inhibitor and kept during the proteasome inhibitor treatment; N-acetyl-cysteine (NAC, Sigma) was dissolved in water and used at 1 and 5 mM. Ascorbic Acid (AA, Sigma) was dissolved in water and used at 200 µM and 500 µM. MitoTempo (Sigma) was dissolved in water and used at 10 and 25 µM. Trolox (Sigma) was dissolved in DMSO and used at 500 µM and 1 mM. The PKR inhibitor (*Zhu et al., 2011*; *Shimazawa and Hara, 2006*) (Cayman) was dissolved in DMSO and used at a final concentration of 1 µM and was added 1 hr before proteasome inhibitor treatment. Protein synthesis was evaluated for all drugs alone and no differences were observed when compared to the control treated samples (data not shown; except for MG-132 and lactacystin). All experiments and dosages were tested in at least two biological rep-licates; the presented quantifications are from the higher dosage tested. Actinomycin-D (Sigma) was dissolved in DMSO and used at 10 µM. It was added at the same time as the PI when indicated.

## Amino acid measurements

1 million neurons (control, or following 30 min or 60 min MG132 20 µM treatment) were collected in 10% SSA (sulfosalicylic acid), centrifuged at maximum speed for 10 min and sent to Zurich ETH Zur-ich/University of Zurich (Functional Genomics Center), for free amino acid analysis measured by reverse phase HPLC.

## Heme measurements

Hippocampal neurons were grown in 10 cm dishes (5 million cells per well), each plate was lysed in 50 µl of the buffer kinase buffer (5 mM MOPS, 2.5 mM β-glycerophosphate, 1 mM EGTA, 0,5 mM EDTA). The mouse hippocampus was lysed in 200 µl of the above buffer. Mouse blood was collected with 3 mM EDTA, the sample was centrifuged 5 min at 1000 rpm to collect the erythrocytes, the cells were extensively washed in PBS, and lysed with kinase buffer. Equal amounts of protein (measured by BCA assay -Pierce) were used for free heme measurement following the manufacturer's instruc-tions (Sigma MAK-036).

## HRI activity in the presence of blood and hippocampal lysates

Blood and hippocampal lysates were obtained as indicated above. To inhibit the endogenous kin-ases and proteases, the lysates were heated at 60°C for 10 min, and centrifuged, 25 µg of the lysates were added to the kinase reaction mix (per reaction: 0.5 µg of recombinant eIF2α, 0.0005–0.001-0.0033–0.0066 µg of recombinant HRI, and 60 µM ATPγ). The reactions were carried out for 45 min at 30°C, stopped with SDS-Page running buffer, run on a gel and transferred to a PVDF membrane that was exposed to a phosphorimager screen.

## Imaging immunocytochemistry and FISH

Images were acquired with a LSM780 confocal microscope (Zeiss) using 20x or 40x objectives (Plan Apochromat 20 x/NA 0.8 M27, Plan Apochromat 40 x/NA 1.4 oil DIC M27). Images were acquired in eight bit mode as Z-stacks with 1024 × 1024 pixels xy resolution through the entire thickness of the cell with optimal overlap of optical z-sections. The detector gain in the signal channel(s) was set to cover the full dynamic range but to avoid saturated pixels. Imaging conditions were held constant within an experiment.

## Image analysis and representation

To quantify puromycin or HA immunoreactivity or ZsGreen fluorescence, three channel maximum intensity projections of the Z-stacks created in the Zeiss Zen software were opened in ImageJ. For single soma analyses all neuronal cell bodies in the image were manually outlined based on Map2 staining (to distinguish from glial cells) and areas saved as ROIs. Hela cell ROIs were outlined based on combined channels selecting a region around the nuclear stain. The image was split into single channel images in ImageJ and the mean grey value for each ROI was measured in the relevant channels. For visualization, maximum intensity projections of the Z-stacks were adjusted for brightness and contrast in ImageJ with the same settings for samples and controls and over the whole presented image. Examples of single dendrites shown were created with the ImageJ Plugin 'Straighten' by tracing a dendrite starting from the soma, resulting in left to right in the image representing proximal to distal.

## Polysome profiling

Cytosolic lysates from primary neuronal cultures were prepared as follows: Cell treated with vehicle or proteasome inhibitors were incubated with 100 µg/ml cycloheximide in the medium for 5 min, washed in cold PBS with 100 µg/ml cycloheximide and lysed in 8% glycerol, 20 mM Tris pH 7.5, 150 mM NaCl, 5 mM MgCl$_2$, 100 µg/ml cycloheximide, 1 mM DTT, 1% Triton X-100, 24 U/ml TURBO DNase, 200 U/mL RNasin(R) Plus RNase Inhibitor. Lysates were pipetted up and down until homogenization was clear with a $0.4 \times 20$ mm needle (HSW FINE-JECT) on ice. Lysates were centrifuged for 10 min at 10,000 g at 4°C, and the supernatants were used for ribosome fractionation.

For sucrose gradients, all solutions were prepared in gradient buffer (20 mM Tris pH 7.5, 8% glycerol, 150 mM NaCl, 5 mM MgCl$_2$, 100 µg/ml cycloheximide, 1 mM DTT). Gradients were prepared by sequentially adding different sucrose concentrations (in order from first added to last, 8 mL of 55%, 0.5 mL of 50%, 0.5 mL of 40%, 0.5 mL of 30%, 0.5 mL of 20%, 0.5 mL of 10%) into the same Thinwall polypropylene tube (Beckman). After the addition of each sucrose solution, tubes were placed at −80°C to freeze the content before the next layer. The gradients were stored at −80°C and left for equilibration at 4°C o.n. Then 0.5 to 1.5 OD (measured with NanoDrop at 260 nm) of the lysates were loaded on top of the gradients and spun at 36,000 rpm at 4°C for 2 hr with a SW41-Ti rotor (Beckman). Fractions from each sample were collected every 9 s using a density gradient fractionation system (Teledyne Isco, intensity used 1), chased by 60% sucrose 10% glycerol in water at 850 µL/min, and continuous monitoring at 254 nm using a UA-6 detector. For quantification of monosome and polysomes peaks, the area under the curve was measured using Image J or the following program: https://software.scic.brain.mpg.de/projects/MPIBR/PolysomeProfiler. The same script was used to visualize the overlay of the polysome profiles of different samples.

## Statistics

Statistical significance of the quantification of IF and WB was tested from single or combined experiments (as stated in the figure legends). If multiple experiments were combined all the data are shown in the same plot. When necessary, data were normalized to one condition. Statistical analysis was performed using GraphPad Prism.

## Acknowledgements

We thank I Bartnik, N Fuerst, A Staab, C Thum and D Vogel for the preparation of primary cell cultures and excellent technical assistance. We thank E Ciirdaeva for her assistance with the mouse colonies. We thank E Northrup, S Zeissler and the animal facility of MPI for Brain Research for their excellent support. We thank Alice Anderton, Ana Wechsler, Dominik Reichert and Niels Hein for their help with some of the experiments. Analysis of rare codons was performed by G Tushev. BA-C was supported by a Marie Curie Intra-European Fellowship for career development, by Ramón y Cajal (MICINN-Spain) and by "Atración de Talento" grants (CAM-Spain). PGD-A is supported by the Peter and Traudl Engelhorn Foundation and the Alexander von Humboldt Foundation (USA-1198990-HFST-P). EMS is funded by the Max Planck Society, DFG CRC 1080: Molecular and Cellular Mechanisms of Neural Homeostasis and DFG CRC 902: Molecular Principles of RNA-based

Regulation and the European Research Council (ERC) under the European Union's Horizon 2020 research and innovation programme (grant agreement No 743216).

## Additional information

### Funding

| Funder | Grant reference number | Author |
|---|---|---|
| Max Planck Society | | Susanne tom Dieck<br>Claudia M Fusco<br>Paul Donlin-Asp<br>Julio D Perez<br>Erin M Schuman |
| European Research Council | | Erin M Schuman |
| Marie Curie Cancer Care | Intra-European Fellowship | Beatriz Alvarez-Castelao |
| Peter and Traudl Engelhorn Foundation | | Paul Donlin-Asp |
| Alexander von Humboldt Foundation | USA-1198990-HFST-P | Paul Donlin-Asp |
| Max Planck Society | DFG CRC 1080 | Erin M Schuman |
| Cellular Mechanisms of Neural Homeostasis | DFG CRC 902 | Erin M Schuman |
| Horizon 2020 - Research and Innovation Framework Programme | 743216 | Erin M Schuman |
| Comunidad de Madrid | Atracción de Talento-2019T1/BMD-14057 | Beatriz Alvarez-Castelao |
| Ministerio de Ciencia e Innovación | Ramón y Cajal- RYC2018-024435-I | Beatriz Alvarez-Castelao |

The funders had no role in study design, data collection and interpretation, or the decision to submit the work for publication.

### Author contributions

Beatriz Alvarez-Castelao, Conceptualization, Data curation, Software, Formal analysis, Validation, Investigation, Visualization, Methodology; Susanne tom Dieck, Claudia M Fusco, Paul Donlin-Asp, Julio D Perez, Data curation, Investigation; Erin M Schuman, Conceptualization, Supervision, Funding acquisition

### Author ORCIDs

Beatriz Alvarez-Castelao (iD) https://orcid.org/0000-0001-7505-1855
Susanne tom Dieck (iD) https://orcid.org/0000-0002-5884-8640
Erin M Schuman (iD) https://orcid.org/0000-0002-7053-1005

### Ethics

Animal experimentation: The housing and sacrificing procedures involving animal treatment and care were conducted in conformity with the institutional guidelines that are in compliance with national and international laws and policies (DIRECTIVE 2010/63/EU; German animal welfare law; FELASA guidelines). The animals were euthanized according to annex 2 of § 2 Abs. 2 Tierschutz-Versuchstier-Verordnung. Animal numbers were reported to the local authority (Regierungspräsidium Darmstadt, approval numbers: V54-19c20/15-F126/1020 and V54-19c20/15-F126/1023).

### Decision letter and Author response

Decision letter https://doi.org/10.7554/eLife.52714.sa1
Author response https://doi.org/10.7554/eLife.52714.sa2

## Additional files

### Supplementary files
• Supplementary file 1. List of rare codons in HRI mRNA.

• Transparent reporting form

### Data availability
All data generated or analysed during this study are included in the manuscript and supporting files.

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
