## [Decision Letter]

**Acceptance summary:**

Little is known regarding the coordination of protein synthesis and protein degradation in neurons. This paper demonstrates that one of the four eIF2α kinases, heme-regulated kinase inhibitor, regulates the decrease in mRNA translation to restore proteostasis during inhibition of the ubiquitin proteasome inhibition system.

**Decision letter after peer review:**

Thank you for submitting your article "Neuronal Proteostasis is mediated by the switch-like expression of Heme-regulated Kinase 1" for consideration by *eLife*. Your article has been reviewed by three peer reviewers, one of whom is a member of our Board of Reviewing Editor, and the evaluation has been overseen by Eve Marder as the Senior Editor.

The reviewers have discussed the reviews with one another and the Reviewing Editor has drafted this decision to help you prepare a revised submission.

Summary:

Alvarez-Castelao et al. present an interesting feedback mechanism to decrease protein synthesis during ubiquitin-proteasome system (UPS) inhibition in neurons. Proteosome inhibition leads to decreased mRNA translation in both neuronal soma and dendrites that was accompanied by eIF2α phosphorylation. Knockout of heme-regulated kinase inhibitor (HRI), but not the other 3 eIF2α kinases, resulted in reduced inhibition of protein synthesis and eIF2α phosphorylation during UPS inhibition. The authors show that basal levels of HRI are low, but increase after UPS inhibition. They propose that the endogenous level of HRI is very low due to degradation by the proteasome. They further suggest that there is also translational regulation of HRI which is due to rare codons in its CDS. Finally, they suggest that the kinase activity of HRI is constitutively high due to low heme levels in neurons.

Essential revisions:

1) It's unexpected and interesting that HRI would be the eIF2α kinase mediating the ISR response in neurons on inhibition of the proteasome. However, it's difficult to accept that only a ~30-40% increase in HRI protein levels is enough to account for the inhibition of bulk protein synthesis by 60 to >90% elicited by PI treatments in different experiments without some other mechanism for activating HRI kinase function, as there are low heme levels regardless of whether or not the proteasome is inhibited. Indeed, there are published results describing mechanisms for controlling HRI activity through dimerization and domain interactions. They should assay the level of phosphorylation of Threonine-485 in HRI, a known site of autophosphorylation in the activation loop and indicator of HRI kinase activity, to determine whether it increases substantially more on PI treatment than the modest 1.3-fold increase seen in HRI protein levels. Alternatively, they could try to show that only a < 2-fold increase in HRI expression in untreated neurons would be sufficient to mimic the effects of PI on eIF2α phosphorylation and general translation, as their model predicts. They show in Figure 6D-E that HRI overexpression has this effect, but do not quantify the extent of overexpression above the native levels.

2) A major branch of the ISR entails the translational activation of transcription factors, notably ATF4, by a re-initiation mechanism involving uORFs, and it is surprising that they have generally ignored this aspect of the response in their experiments, with only one experiment in Figure 5—figure supplement 1A showing increased polysome association of ATF4 mRNA on PI treatment. Given that the quality of the Western blots used to quantify eIF2α phosphorylation is not very high, and the multiple WB quantifications of peIF2α and unphosphorylated eIF2α needed to determine ratios of peIF2α:total eIF2α, it would be very reassuring to show that polysome association of ATF4 mRNA, or expression of an ATF4 reporter, increases on PI treatment of WT but not of HRI KO cells. Alternatively, qRT-PCR for ATF4 targets could be performed under various conditions.

3) There are a number of experiments in which treatments are found not to affect the translational response to PI, but no controls are presented for these negative results to convince us that the treatments had the intended effects in their hands, on their cells. In other cases, controls are lacking to establish the specificities of phospho-specific antibodies, or to show that they achieved a clean separation of dendrites from cell bodies. These controls have to be included for the results to be interpretable. These include the following:

– Controls are required to document that the dendrite preparation in Figure 1H is not contaminated with cell bodies.

– The experiment in Figure 2—figure supplement 1 should be described explicitly in the text of Results, and controls are required to document that actinomycin treatment successfully blocked transcription.

– For Figure 2—figure supplement 1C, controls are required to document the specificity of the phospho-specific antibodies and the ability to observe changes in phosphorylation using them in response to, eg. rapamycin treatment of their cell lines.

– For Figure 3A, it is necessary to indicate the nature of the PKR inhibitor and to document that it was effective at inhibiting PKR activity in the neurons.

– Controls are lacking for the experiments in Figure 5—figure supplement 2 to assure us that the treatments used to quench ROS, block NO synthesis were effective in their cells/hands. The same goes for the heme oxygenase inhibitors; although in this case, the logic of the experiment also appears to be flawed, as inhibitors of heme breakdown should lead to higher, not lower, heme levels and thus prevent, not elicit, HRI activation.

4) The results in Figure 6B could be explained differently by proposing that the hippocampal lysate has lower levels of peIF2α phosphatase. This alternative explanation could be ruled out by incubating peIF2α in each lysate (without added HRI) and measuring rates of dephosphorylation over time. Alternatively, they could show that similar levels of peIF2α are produced in both lysates when Gcn2, PERK or PKR are added instead of HRI.

– What was the loading control used for Figure 4A and 4B quantifications? Also there are no loading controls for the representative western shown in Figure 4—figure supplement 1D.

5) The results in Figure 5—figure supplement 1C contradict the claim that codon optimization increases polysome association of HRI-HA mRNA claimed from the data in Figure 5—figure supplement 1B, as the wt and opt mRNAs show the same polysome association in control conditions in Figure 5—figure supplement 1C. This discrepancy needs to be addressed in the text.

6) Regarding Figure 6C, it should be verified that hemin does not stimulate protein synthesis when added to HRI KO cells treated with PI; and that hemin addition suppresses the increase in peIF2α elicited by PI.

7) The statement in the Discussion that the data indicate, among other findings, that "iii) the initial decrease in general translation leads to an increase in the availability of rare codons (Saikia et al., 2016), further increasing the translation of HRI mRNA" is a plausible scenario but in no way demonstrated by the data. Also, I'm not sure what they mean by availability of rare codons-probably they mean the cognate tRNAs for such codons.

8) Related to point 1, the statements in the Discussion "Our data indicate that above multiple mechanisms cleverly cooperate to enhance the activity and expression of HRI in response to reduced proteasome activity. These mechanisms make HRI an optimized sensor and effector for sensing and responding to proteasome inhibition." seem premature, as they have identified no mechanism for enhancing HRI activity, although one might well exist that remains to be uncovered, and document only a small increase in HRI expression which may or may not be adequate to explain the observed translational responses.

9) The authors examined HRI-HA expression in Figure 4E-G and found it much lower than the ZsGreen reporter expressed by the DOX-inducible bi-directional promoter. Is it possible that the HA tag destabilized HRI? Figure 4F-G: It is not specified how long the Doxycycline treatment was – this should be added. The experiment could be repeated treating with Doxycycline for 1 h to test whether the short half-life of HRI contributes to the lack of detection. Is it possible that there is directionality of the promoter in this situation (i.e., were the cDNAs switched relative to the promoter to see if similar results were obtained?

10) Figure 4—figure supplement 1B: The loading control is missing and should be added. The figure legend states that the related analysis is in Figure 4D – do the authors mean Figure 4B? It would be clearer to move the representative image (Figure 4—figure supplement 1B) close by its related plot (Figure 4B?).

11) Figure 4—figure supplement 1E: The plot should show also the quantification of the condition "+ PI".

12) Figure 5—figure supplement 2H: Statistical analyses between the PI and the PI+ hemin conditions are missing. Figure 5—figure supplement 2H does not seem to be referenced in the manuscript. It is unclear whether Figure 5—figure supplement 2H and Figure 6C are the same experiment? If redundant, it should be removed.

13) Related to point 1, the statements in the Discussion "Our data indicate that above multiple mechanisms cleverly cooperate to enhance the activity and expression of HRI in response to reduced proteasome activity. These mechanisms make HRI an optimized sensor and effector for sensing and responding to proteasome inhibition." seem premature, as they have identified no mechanism for enhancing HRI activity, although one might well exist that remains to be uncovered, and document only a small increase in HRI expression which may or may not be adequate to explain the observed translational responses.

14) In addition to protein expression, does hemin also rescue eIF2α phosphorylation increases in PI conditions?

[Editors' note: further revisions were suggested prior to acceptance, as described below.]

Thank you for resubmitting your work entitled "The switch-like expression of Heme-regulated kinase 1 mediates neuronal proteostasis following proteasome inhibition" for further consideration by *eLife*. Your revised article has been evaluated by Timothy Behrens (Senior Editor) and a Reviewing Editor.

The manuscript has been improved but there are some remaining issues that need to be addressed it before acceptance. These largely include changes to the text including adding citations, and inclusion of data as a supplemental figure and are outlined by reviewer 3 below.

Reviewer #2:

The authors have addressed my comments well. The revised manuscript presents a biologically significant set of findings that bring novel insights into proteostasis during proteasome inhibition and, in my view, is very worthy of publication.

Reviewer #3:

The revised paper has been improved and the additional information provided in the rebuttal serve adequately to allay my major concerns. There are however some additional revisions to the text that should be made to improve the quality of the report, as described below:

Essential Revision point 1. Regarding the data provided in the rebuttal and described as "Below we plot our imaging data (from Figure 4E-G of the original and revised manuscript) measuring the level of HRI expression (using a transfected construct) and the corresponding nascent protein signal (obtained using brief puromycin labeling) in the same neurons. As expected, [.…]. Consequently, it is possible that a 30% increase in HRI expression decreases protein synthesis by more than 50%." These new experimental data and analysis should be provided in a supplementary figure and summarized in the Results to bolster the key conclusion that only a 30% increase in HRI expression could potentially decrease protein synthesis by more than 50%.

Essential Revision point 3: the controls provided for the reviewer's benefit should be included as supplementary data, as they will be valuable for all readers of the paper. Note however that the control data provided to validate the antibodies against phosphorylated 4EBP are hard to evaluate. Should all of the phosphorylated isoforms be reduced equally by rapamycin? Also, an immunofluorescence assay was provided that was labeled incorrectly as an immunoblot and is difficult to evaluate without quantification and explanation. They should cite the literature justifying the use of the PKR inhibitor without confirming its efficacy in their hands/cells.

Essential Revision point 4: The legend to Figure 6B should be modified to reflect the use of heat inactivated extracts. Also, the conditions for the assays with purified components should be included in the Materials and methods, and the legend to Figure 6A should indicate the use and concentration of purified components. The legends to Figure 4A-B should be revised to indicate the method for normalization of the data (Figure 4A) and to indicate that co-immunoprecipited samples were quantified (Figure 4B).

Essential Revision point 7: The sentence in the Discussion was modified, but it would be even more accurate to say that the availability of charged tRNAs was being altered, to distinguish this from alterations in tRNA abundance which would be unexpected. Also, in the Abstract it'd be more correct to state "Following proteasome inhibition, HRI protein levels increase owing to stabilization of HRI and enhanced translation presumably via the increased availability of charged tRNAs for its rare codons."

---

## [Author Response]

Essential revisions:1) It's unexpected and interesting that HRI would be the eIF2α kinase mediating the ISR response in neurons on inhibition of the proteasome. However, it's difficult to accept that only a ~30-40% increase in HRI protein levels is enough to account for the inhibition of bulk protein synthesis by 60 to >90% elicited by PI treatments in different experiments without some other mechanism for activating HRI kinase function, as there are low heme levels regardless of whether or not the proteasome is inhibited. Indeed, there are published results describing mechanisms for controlling HRI activity through dimerization and domain interactions. They should assay the level of phosphorylation of Threonine-485 in HRI, a known site of autophosphorylation in the activation loop and indicator of HRI kinase activity, to determine whether it increases substantially more on PI treatment than the modest 1.3-fold increase seen in HRI protein levels.

We agree that there are mechanisms besides heme reduction capable of changing the HRI activity, we have tested most of the mechanisms described thus far and found no effect on the translational response mediated by HRI activity in our experimental system.

Unfortunately, it is not possible to check HRI threonine-485 phosphorylation in neurons due to the very low expression of HRI, and the lack of good antibodies. In liver and blood, where HRI has been studied more extensively, the high HRI protein levels make it much easier to study and we can detect the hyperphosphorylated version in blood ourselves (see Figure 4—figure supplement 1B). The study of this modification by mass spectrometry is also not possible because we failed to identify HRI by MS. We must emphasize that a 1.3 fold increase in an active kinase cannot be qualified as modest, for example the specific activity of recombinant HRI is exponentially related to the amount of protein until it gets saturated (see for example the specific activity assay for the purified HRI from Sigma-Aldrich (https://www.sigmaaldrich.com/catalog/product/sigma/srp5321?lang=es&region=ES&gclid=EAIaIQobChMIyITW2e6N6QIVR9HeCh1aXArbEAAYASAAEgLOdfD_BwE).

Importantly, in cells eIF2B levels are limiting, for this reason partial eIF2α phosphorylation in vivo is able to fully repress eIF2B function and protein synthesis initiation (Adomavicius et al., 2019).

Alternatively, they could try to show that only a < 2-fold increase in HRI expression in untreated neurons would be sufficient to mimic the effects of PI on eIF2α phosphorylation and general translation, as their model predicts. They show in Figure 6D-E that HRI overexpression has this effect, but do not quantify the extent of overexpression above the native levels.

Thanks for this suggestion- we see that it is worthwhile to really explain how small changes in kinase activity can have a big effect on global protein synthesis levels. We plot our imaging data (from figure 4E-G of the original and revised manuscript) measuring the level of HRI expression (using a transfected construct) and the corresponding nascent protein signal (obtained using brief puromycin labeling) in the same neurons. As expected, the correlation between the expression of HRI kinase and the inhibition in protein synthesis is not linear, the best fit corresponds to a power line. This is consistent with the assumption that one molecule of HRI will lead to the phosphorylation of multiple molecules of substrate. Resulting from this form of dependence, small increases in HRI expression will exert most impact on protein synthesis within a low to intermediate range: when expression is not too low to impact eIF2α phosphorylation, but also not too high because then already most protein synthesis is shut off. Consequently, it is possible that a 30% increase in HRI expression decreases protein synthesis by more than 50%.

2) A major branch of the ISR entails the translational activation of transcription factors, notably ATF4, by a re-initiation mechanism involving uORFs, and it is surprising that they have generally ignored this aspect of the response in their experiments, with only one experiment in Figure 5—figure supplement 1A showing increased polysome association of ATF4 mRNA on PI treatment. Given that the quality of the Western blots used to quantify eIF2α phosphorylation is not very high, and the multiple WB quantifications of peIF2α and unphosphorylated eIF2α needed to determine ratios of peIF2α:total eIF2α, it would be very reassuring to show that polysome association of ATF4 mRNA, or expression of an ATF4 reporter, increases on PI treatment of WT but not of HRI KO cells. Alternatively, qRT-PCR for ATF4 targets could be performed under various conditions.

We are a bit stung by the criticism that the Western Blot quality is not high- this was an extremely challenging experiment. eIF2α phosphorylation is very labile- despite this, we obtained well defined bands, and we always loaded in the same gel biological replicates, in addition to the experimental replicates. We don’t think that ATF4 is a good additional readout, because other pathways can activate its translation, for example mTOR (Park et al., 2017) and the ATF4 protein is itself degraded by the proteasome (Lassot et al., 2001). The ratio of eIF2α vs peIF2α is the most accurate way to measure peIF2α changes, whereas measuring ATF4 or its target genes is indirect and can lead to misleading conclusions.

3) There are a number of experiments in which treatments are found not to affect the translational response to PI, but no controls are presented for these negative results to convince us that the treatments had the intended effects in their hands, on their cells. In other cases, controls are lacking to establish the specificities of phospho-specific antibodies, or to show that they achieved a clean separation of dendrites from cell bodies. These controls have to be included for the results to be interpretable. These include the following:– Controls are required to document that the dendrite preparation in Figure 1H is not contaminated with cell bodies.

We prepare neuronal cultures on these filters routinely in the laboratory, and we perform several tests to prove the quality of the separation of the neuronal compartments, which is always very good. We now refer in the revised manuscript to our recent publication (Biever et al., 2020) (see Figure S1H which illustrates the quality of the separation). In addition, the Author response image 1 shows that Lamin B1 a protein confined to the neuronal somata is indeed mainly found in the cell material on the upper surface of the filters.

**Author response image 1. sa2fig1:** Western Immunoblot for Lamin B1. Neurons were scraped from the upper and lower part of the membrane and the nuclear protein Lamin B1 was detected by Western Blot.

– The experiment in Figure 2—figure supplement 1 should be described explicitly in the text of the Results, and controls are required to document that actinomycin treatment successfully blocked transcription.

We performed an experiment and prepared Author response image 2 for the reviewers which shows that ActD works as expected in our experimental system but did not include it into the revised manuscript.

**Author response image 2. sa2fig2:** Testing actinomycin D’s effectiveness in blocking transcription using ddPCR of Arc mRNA. Neurons were treated with ActD for 2h and mRNA levels of Arc, an immediate early gene, in non-treated and treated neurons were measured. In the left graph the ddPCR counts for Arc mRNA and ERCC (used for normalization) are shown. Treatment with actinomycin D (indicated by “+”) completely prevented the transcription of Arc mRNA. In the right graph, the average of the experiments normalized to the ERCC standards are shown.

– For Figure 2—figure supplement 1C, controls are required to document the specificity of the phospho-specific antibodies and the ability to observe changes in phosphorylation using them in response to, eg. rapamycin treatment of their cell lines.

We do not use cell lines but primary neurons for our experiments. The phospho-specific antibodies used for this paper were already tested using Rapamycin by the commercial supplier (see Cell-Signaling website for the different antibodies; peIF4G- https://www.cellsignal.com/products/primary-antibodies/phospho-eif4g-ser1108-antibody/2441?site-search-type=Products; p4EBP- https://www.cellsignal.com/products/primary-antibodies/phospho-4e-bp1-thr37-46-236b4-rabbit-mab/2855?site-search-type=Products; peIF4B- https://www.cellsignal.com/products/primary-antibodies/phospho-eif4b-ser406-d1c10-rabbit-mab/8151?site-search-type=Products) and are widely used by research groups in the field. Nevertheless, we treated neurons with Rapamycin for 2h and confirmed the specificity of the antibodies (see Author response image 3).

**Author response image 3. sa2fig3:** Western Immunoblot for phosphorylated translation factors. Neurons were treated with Rapamycin (10 µm) for 2h and the phosphorylation of 4EBP1, EIF4G and EIF4B was detected by Western Blot using the phospho-specific antibodies described in the paper. A decrease in the phosphorylation of the proteins in response to Rapamycin treatment was observed.

– For Figure 3A, it is necessary to indicate the nature of the PKR inhibitor and to document that it was effective at inhibiting PKR activity in the neurons.

The PKR Inhibitor is an oxindole/imidazole derivative that binds the ATP-binding site of PKR and blocks autophosphorylation with an IC50 value of 186-210 nM (Zhu et al., 2011). We used a concentration of 1µM for 4h. The inhibitor is so powerful that it even works in vivo by intraperitoneal administration reducing the phosphorylation of PKR and eIF2α in the brain (Ingrand et al., 2007). Similar administration in mice enhances long-term memory storage, including contextual and auditory long-term fear memories (Zhu et al., 2011). Given the profound effect of the HRI KO on protein synthesis levels after brief proteasome inhibition, it is clear that HRI is the primary kinase activated in response to brief proteasome inhibition in our experiments. Of course, we cannot rule out the idea that other kinases could also contribute to eIF2α phosphorylation during prolonged proteasome inhibition, for example. Indeed, we include mention of this possibility in our Discussion.

– Controls are lacking for the experiments in Figure 5—figure supplement 2 to assure us that the treatments used to quench ROS, block NO synthesis were effective in their cells/hands. The same goes for the heme oxygenase inhibitors; although in this case, the logic of the experiment also appears to be flawed, as inhibitors of heme breakdown should lead to higher, not lower, heme levels and thus prevent, not elicit, HRI activation.

A useful and robust pharmacological strategy for testing whether a particular pathway is involved in a cellular process is the use of several different inhibitors. This is the approach we have taken. We used several (3 different ROS quenchers, 4 different NOS inhibitors and 3 different HO inhibitors) chemically distinct well-characterized inhibitors (used in hundreds of published studies) acting on the same pathway, and several dosages (data not shown) of each of these inhibitors. It is extremely unlikely that none of the drugs were working at any tested dosage. Additionally, we have measured ROS and NO species directly and we failed to detect any changes after proteasome inhibition (data not shown). We note that the direct measurement of Heme Oxygenase activity is extremely complicated and has low sensitivity and therefore especially challenging in primary cultured neurons. We don’t believe the logic of our Heme oxygenase inhibitor experiment is flawed. In fact, we were looking for an inhibition of HRI activity using the heme breakdown inhibitors. The rationale for the use of Heme oxygenase inhibitors was to test the possibility that proteasome inhibition could activate Heme breakdown, via activation of the Heme oxygenase. We thus used the HO inhibitors as a tool to block heme breakdown and block any resulting activation of HRI. We did not observe any rescue of protein synthesis inhibition when any of the HO inhibitors were added together with the proteasome inhibitor, thus we can conclude that heme breakdown mediated by heme oxygenase does not participate in the global downregulation of protein synthesis.

4) The results in Figure 6B could be explained differently by proposing that the hippocampal lysate has lower levels of peIF2α phosphatase. This alternative explanation could be ruled out by incubating peIF2α in each lysate (without added HRI) and measuring rates of dephosphorylation over time. Alternatively, they could show that similar levels of peIF2α are produced in both lysates when Gcn2, PERK or PKR are added instead of HRI.

Yes, we agree with the idea the reviewer presents. Indeed, to avoid artifacts due to other enzymatic activities than the one we were analyzing (HRI activity), all the enzymatic activities from the lysate were inactivated by denaturation (heating up the samples as indicated in the Materials and methods section). The cleared lysates thus contained only soluble material such as free Heme prior to adding the purified HRI and eIF2α to the lysates to perform the activity assay.

– What was the loading control used for Figure 4 a and 4B quantifications? Also there are no loading controls for the representative western shown in Figure 4—figure supplement 1D.

Figure 4A is a ddPCR experiment where levels of HRI were normalized to ribosomal RNA. The experiments represented in Figure 4B and Figure 3—figure supplement 1D are immunoprecipitation experiments. In these experiments, one specific antibody is used to isolate (pull-down) the protein of interest (POI). The POI remains bound to the antibody- the antibody is bound to a physical support such as magnetic beads. The beads+antibody are used as bait to isolate the POI from the rest of the proteins of the tissue. The beads+antibody+POI are loaded onto a gel with the aim of quantifying the amount of protein present in the lysate. The nature of the technique makes it not suitable for loading controls, as other proteins (different from the POI) that could serve as “loading controls” will be in the immunoprecipitates owing to non-specific binding to the beads or to the antibody, consequently its presence in the lysate is non-specific and potentially variable and hence cannot be used for quantification. The way to make IPs semi-quantitative is repeating them several times (and of course loading the same amount of total protein to each IP by measuring the protein used for the input by an accurate method- in our case QBIT). The data represented in the figure come from 4 independent experiments, each of them was performed with 6 separate dishes of neurons. Additionally, we present a negative control IP from the HRI knockout neurons. The IPs are done with an excess of the antibody (we calculated the amount of antibody necessary by titration experiments), to ensure the precipitation of the total amount of endogenous HRI and prevent saturation of the antibody.

We did this experiment by IP, because we cannot detect HRI in the intact lysates. In addition, the increased HRI expression demonstrated by the IP experiments is supported by the measurement of the translation rates of HRI by ddPCR and Polysome Profiling, and by the experiment demonstrating that HRI is degraded by the proteasome. Both processes contribute to the increase in expression of HRI shown by the IP experiments.

5) The results in Figure 5—figure supplement 1C contradict the claim that codon optimization increases polysome association of HRI-HA mRNA claimed from the data in Figure 5—figure supplement 1B, as the wt and opt mRNAs show the same polysome association in control conditions in Figure 5—figure supplement 1C. This discrepancy needs to be addressed in the text.

There is no discrepancy. Please note that the normalization in the two graphs is different to emphasize different findings from the same experiment. In the Figure 5—figure supplement 1B we show the higher association to the polysomes of the optimized HRI mRNA in control conditions, and in the Figure 5—figure supplement 1C the change in the behavior of the optimized HRI mRNA in comparison to the nonoptimized HRI mRNA after proteasome inhibition, for this reason both control conditions have the value of 1. To make the figure easier to interpret we have now split Figure 5—figure supplement 1C in two graphs and added a description to the legend on how the normalization was conducted.

6) Regarding Figure 6C, it should be verified that hemin does not stimulate protein synthesis when added to HRI KO cells treated with PI; and that hemin addition suppresses the increase in peIF2α elicited by PI.

The reviewer is concerned about potential off-target effects of hemin in the absence of HRI. To address this, we added hemin to HRI KO primary neurons. We did not observe significant changes in basal protein synthesis due to the addition of hemin (Figure 5—figure supplement 1 I-J). Furthermore, the addition of hemin+PI in wt neurons (expressing HRI that can actually respond to heme), showed a significant reduction of the phosphorylation of eIF2α (Figure 5—figure supplement 1 K-L), but not a complete blockade. This is consistent with the results shown in the HRI KO neurons for eIF2α phosphorylation (Figure 3F), and translation (Figure 3C).

7) The statement in the Discussion that the data indicate, among other findings, that "iii) the initial decrease in general translation leads to an increase in the availability of rare codons (Saikia et al., 2016), further increasing the translation of HRI mRNA" is a plausible scenario but in no way demonstrated by the data. Also, I'm not sure what they mean by availability of rare codons-probably they mean the cognate tRNAs for such codons.

We have changed this sentence.

8) Related to point 1, the statements in the Discussion "Our data indicate that above multiple mechanisms cleverly cooperate to enhance the activity and expression of HRI in response to reduced proteasome activity. These mechanisms make HRI an optimized sensor and effector for sensing and responding to proteasome inhibition." seem premature, as they have identified no mechanism for enhancing HRI activity, although one might well exist that remains to be uncovered, and document only a small increase in HRI expression which may or may not be adequate to explain the observed translational responses.

We disagree. We have identified a mechanism for enhancing the activity of HRI- by increasing its expression. We show that in neurons heme levels are very low, leading to a constitutive activation of the kinase. We have now changed this line to indicate “enhance the expression and activity”.

9) The authors examined HRI-HA expression in Figure 4E-G and found it much lower than the ZsGreen reporter expressed by the DOX-inducible bi-directional promoter. Is it possible that the HA tag destabilized HRI? Figure 4F-G: It is not specified how long the Doxycycline treatment was – this should be added. The experiment could be repeated treating with Doxycycline for 1 h to test whether the short half-life of HRI contributes to the lack of detection. Is it possible that there is directionality of the promoter in this situation (i.e., were the cDNAs switched relative to the promoter to see if similar results were obtained?

We agree with the reviewer that sometimes adding a tag to a protein and the position of the tag can change its half-life. In our initial experiments we tried several different induction times with Doxycycline with the original construct carrying ZsGreen and HRI-HA on the same plasmid. We never saw measurable amounts of HRI-HA expression and we set out to investigate the reviewers’ question. Instead of switching the order relative to the promoter we 1) placed the cDNAs on two different plasmids (to lower the amount of ZsGreen and to carry out controls without ZsGreen expression), 2) compared the expression with the HA tag placed on the N-terminus vs attachment to the C-terminus and 3) compared codon-optimized versions of both N- and C-terminally tagged HRI with the wild-type non-optimized tagged HRI.

From these experiments we were able to draw the following conclusions:

With the codon-optimized tagged versions of HRI we obtained measurable expression with long (16-24 h) Dox induction (e.g. data shown in Figure 5E-G). The resulting protein from the codon-optimized version is identical to the protein from the wild-type non-optimized version therefore destabilization by the HA tag can’t be a major explanation for the difficulty to express HRI without the tag. Rather the codon-usage in the wild-type sequence plays a major role.

Furthermore, we investigated the influence of the degradational component on HRI expression by short time (2hr) induction of either C-terminally or N-terminally tagged HRI expression in absence or presence of MG132 (see Author response image 4). Proteasome inhibition increases (in accordance with our model that HRI is a proteasome substrate) the expression levels of both N- and C-terminally tagged HRI to a roughly equal extent (see Author response image 4).

**Author response image 4. sa2fig4:** Expression of transfected HRI in primary neurons -/+ proteasome inhibition, after short induction with Doxycycline (2h). Left panel shows the expression of HRI tagged on the N- terminus, and the right panel shows the expression of HRI tagged on the C-terminus. HRI expression was measured by Immunofluorescence using the HA tag. tf = transfected, PI = proteasome inhibitor.

Taken together, the experiments confirm our two proposed mechanisms to keep HRI protein levels low in neurons: suboptimal translation rapid degradation by the proteasome and reduced synthesis, due at least impart to codon nonoptimality.

Although we never observed significant expression of HRI construct (above background levels- see Author response image 4) with 2h Doxycycline induction, we carried out the reviewer’s requested experiment with 1h Doxycycline induction with the bidirectional expression plasmid. As expected, we did not observe measurable HRI-HA expression- it is simply too short of an induction for this particular protein.

10) Figure 4—figure supplement 1B: The loading control is missing and should be added.

There is a loading control already in this figure. If the reviewer is referring to panel D and not B, please refer to the answer to this same question in point 4.

The figure legend states that the related analysis is in Figure 4D – do the authors mean Figure 4B?

We corrected this typo in the text.

It would be clearer to move the representative image (Figure 4—figure supplement 1B) close by its related plot (Figure 4B?).

We prefer to leave this as is, the quantification is the most meaningful part of this specific experiment.

11) Figure 4—figure supplement 1E: The plot should show also the quantification of the condition "+ PI".

This condition is already taken into account in the represented results, we have modified the figure legend to clarify this point.

12) Figure 5—figure supplement 2H: Statistical analyses between the PI and the PI+ hemin conditions are missing. Figure 5—figure supplement 2H does not seem to be referenced in the manuscript. It is unclear whether Figure 5—figure supplement 2H and Figure 6C are the same experiment? If redundant, it should be removed.

The experiment in the Figure 6C was done by western blot and the one in Figure 5—figure supplement 2H was done in situ, we clarified this in the figure legends and added the missing figure reference in the text.

13) Related to point 1, the statements in the Discussion "Our data indicate that above multiple mechanisms cleverly cooperate to enhance the activity and expression of HRI in response to reduced proteasome activity. These mechanisms make HRI an optimized sensor and effector for sensing and responding to proteasome inhibition." seem premature, as they have identified no mechanism for enhancing HRI activity, although one might well exist that remains to be uncovered, and document only a small increase in HRI expression which may or may not be adequate to explain the observed translational responses.

Repeat of comment 8- see answer above.

14) In addition to protein expression, does hemin also rescue eIF2α phosphorylation increases in PI conditions?

Repeat of comment 6-see answer above.

[Editors' note: further revisions were suggested prior to acceptance, as described below.]

The manuscript has been improved but there are some remaining issues that need to be addressed it before acceptance. These largely include changes to the text including adding citations, and inclusion of data as a supplemental figure and are outlined by Reviewer 3 below.Reviewer #3:The revised paper has been improved and the additional information provided in the rebuttal serve adequately to allay my major concerns. There are however some additional revisions to the text that should be made to improve the quality of the report, as described below:Essential Revision point 1. Regarding the data provided in the rebuttal and described as "Below we plot our imaging data (from Figure 4E-G of the original and revised manuscript) measuring the level of HRI expression (using a transfected construct) and the corresponding nascent protein signal (obtained using brief puromycin labeling) in the same neurons. As expected, [.…]. Consequently, it is possible that a 30% increase in HRI expression decreases protein synthesis by more than 50%." These new experimental data and analysis should be provided in a supplementary figure and summarized in the Results to bolster the key conclusion that only a 30% increase in HRI expression could potentially decrease protein synthesis by more than 50%.

We added a new supplementary figure (Figure 6—figure supplement 1).

Essential Revision point 3: the controls provided for the reviewer's benefit should be included as supplementary data, as they will be valuable for all readers of the paper. Note however that the control data provided to validate the antibodies against phosphorylated 4EBP are hard to evaluate. Should all of the phosphorylated isoforms be reduced equally by rapamycin? Also, an immunofluorescence assay was provided that was labeled incorrectly as an immunoblot and is difficult to evaluate without quantification and explanation. They should cite the literature justifying the use of the PKR inhibitor without confirming its efficacy in their hands/cells.

We added a panel with the controls for the antibodies in the Figure 2—figure supplement 1. The 4EBP1 antibody has more than 700 references on the Cell Signaling web page, supporting its specificity. The 4EBP1 phosphorylated isoforms that can be detected depend on the type of gels that each lab runs. The best way to detect “all” the phosphorylated isoforms is to use a bi-dimensional gel, otherwise one can only see a global reduction in the phosphorylated isoforms as we show here. Additionally, we used an antibody that is specific for the phosphorylation of a specific Threonine, the co-occurrence of the phosphorylation in these positions with other phosphorylation sites is not within the scope of this paper. This antibody was chosen because it labels the most common phosphorylation positions in 4eBP1.

Essential Revision point 4: The legend to Figure 6B should be modified to reflect the use of heat inactivated extracts. Also, the conditions for the assays with purified components should be included in the Materials and methods, and the legend to Figure 6A should indicate the use and concentration of purified components. The legends to Figure 4A-B should be revised to indicate the method for normalization of the data (Figure 4A) and to indicate that co-immunoprecipited samples were quantified (Figure 4B).

We modified the figure legends and the Materials and methods as requested.

Essential Revision point 7: The sentence in the Discussion was modified, but it would be even more accurate to say that the availability of charged tRNAs was being altered, to distinguish this from alterations in tRNA abundance which would be unexpected. Also in the Abstract it'd be more correct to state "Following proteasome inhibition, HRI protein levels increase owing to stabilization of HRI and enhanced translation presumably via the increased availability of charged tRNAs for its rare codons."

We modified the Abstract. We added “charged tRNAs” in the Discussion section.